# MULTIPLE HEADS ARE BETTER THAN ONE: MIXTURE OF MODALITY KNOWLEDGE EXPERTS FOR ENTITY REPRESENTATION LEARNING

**Yichi Zhang**[1,2], **Zhuo Chen**[1,2], **Lingbing Guo**[1,2], **Yajing Xu**[1,2], **Binbin Hu**[3], **Ziqi Liu**[3]
**Wen Zhang**[4,2]*, **Huajun Chen**[1,2,5]*
[1]College of Computer Science and Technology, Zhejiang University
[2]ZJU-Ant Group Joint Lab of Knowledge Graph [3]Ant Group
[4]School of Software Technology, Zhejiang University
[5]Zhejiang Key Laboratory of Big Data Intelligent Computing
`{zhangyichi2022, zhang.wen, huajunsir}@zju.edu.cn`

## ABSTRACT

Learning high-quality multi-modal entity representations is an important goal of multi-modal knowledge graph (MMKG) representation learning, which can enhance reasoning tasks within the MMKGs, such as MMKG completion (MMKGC). The main challenge is to collaboratively model the structural information concealed in massive triples and the multi-modal features of the entities. Existing methods focus on crafting elegant entity-wise multi-modal fusion strategies, yet they overlook the utilization of multi-perspective features concealed within the modalities under diverse relational contexts. To address this issue, we introduce a novel framework with **M**ixture **o**f **Mo**dality **K**nowledge experts (MoMoK for short) to learn adaptive multi-modal entity representations for better MMKGC. We design relation-guided modality knowledge experts to acquire relation-aware modality embeddings and integrate the predictions from multi-modalities to achieve joint decisions. Additionally, we disentangle the experts by minimizing their mutual information. Experiments on four public MMKG benchmarks demonstrate the outstanding performance of MoMoK under complex scenarios. Our code and data are available at `https://github.com/zjukg/MoMoK`.

## 1 INTRODUCTION

Multi-modal knowledge graphs (MMKGs) (Chen et al., 2024) are an extension of traditional knowledge graphs (KGs) (Liang et al., 2024b), encompassing rich modality information such as images and textual descriptions of large-scale entities, which bridges structured knowledge triple and unstructured multi-modal content together. Based on such a specialized and expressive data organization format, MMKGs have evolved the emerging infrastructure of Artificial Intelligence (AI), contributing to numerous AI-related fields like large language models (Zhang et al., 2023b), recommendation systems (Wang et al., 2019a), and other practical applications (Chen et al., 2023a; Liang et al., 2024a).

Learning better entity representation is a crucial topic in the representation learning and KG community. It aims to encode the complex information in the given KG to perform KG reasoning tasks like knowledge graph completion (KGC) (Liang et al., 2023), which is an interesting and important task seeking to automatically discover new knowledge from the existing KGs. KGC allows for the missing entity prediction to a given entity-relation query, e.g., *(ICLR 2025, Located In, ?)*. Traditional KGC usually learn entity representations by modeling the triple structure in the embedding space. As for MMKG, the situation becomes more complex. Multi-modal knowledge graph completion (MMKGC) further enhances the entity embeddings with multi-modal features, aiming to collaboratively model the triple structure and multi-modal content to achieve robust prediction.

Existing MMKGC methods (Wang et al., 2021; Xu et al., 2023) typically employ a multi-modal fusion module to integrate the information from different modalities to obtain joint entity em-

---

*Corresponding Authors.

beddings. These entity embeddings are then mapped into a scalar score along with the relation embeddings as a basis for assessing the triple plausibility. MMKGC, being a prediction task in a multi-relational scenario, is influenced by different relational contexts, which in turn affect the selection and utilization of entity modality features. As illustrated in Figure 1, different sections of varied modality information emphasize their respective significance when making predictions based on different relationships. However, such a conventional paradigm overlooks the information diversity both inter-modality and intra-modality. Different modalities can represent various aspects of entity information, and information within the same modality can also play different roles depending on the relational context. If vanilla multi-modal fusion is performed directly at the entity level without considering the relational context, it can result in low utilization of this multi-modal information and finally learn immutable entity embeddings across different relational contexts, thereby limiting the model's performance. This limitation is particularly pronounced in realistic scenarios where many entities are subject to modal noise and incomplete information, which can further challenge the model's ability to utilize modal information effectively.

To address these issues, we propose a **M**ixture **o**f **Mo**dality **K**nowledge experts (MoMoK) framework in this paper. MoMoK incorporates relation-guided modality knowledge experts for each modality, which constructs expert networks in each modality. These expert networks, guided by the relational context of the current triple, adaptively aggregate the multi-view embeddings for entities. Further, MoMoK employs multimodal joint decision to integrate the modality

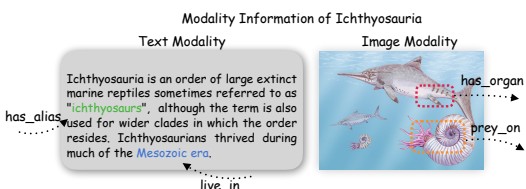

Figure 1: Different relational context requires different modality information for proper prediction.

embeddings as a new joint modality and achieve comprehensive triple prediction in an ensemble manner. Ultimately, we employ an expert information disentanglement module to differentiate learning across different expert networks with contrastive mutual information estimation, aiming to force different experts to specialize in different relational contexts. This entire process can be likened to each **modality functioning as a senior expert, gathering the insights of junior experts within the corresponding modality**, based on the principle that multiple heads are better than one. These insights are then communicated and integrated across modalities to facilitate more comprehensive decision-making. We conduct comprehensive experiments on four public MMKG benchmarks to demonstrate the effectiveness of our framework MoMoK with further exploration to validate its properties. Our contribution can be summarized as:

- We address the problems in modality information utilization by MMKGC models and propose MoMoK with relational-guided modality experts and multi-modal joint decision to learn better entity representations and unleash the power of multi-modal information in MMKGs.

- We examine the learning of different modal experts through the lens of mutual information estimation, and propose to decouple and discretize the expert information within a modality using mutual information comparison estimation. We present detailed theoretical analysis to justify the design.

- We conduct extensive experiments against 20 recent baselines on four MMKG benchmarks to demonstrate the state-of-the-art (SOTA) performance of MoMoK and further explore its robustness, reasonability, and interpretability in the complex environments.

## 2 RELATED WORKS

**Multi-modal Knowledge Graph Completion (MMKGC)** MMKGC (Chen et al., 2024) aims to automatically discover new knowledge triples from the existing MMKGs by collaboratively modeling the triple structure and multi-modal information (e.g., images and textual descriptions) in the MMKGs. Mainstream MMKGC methods (Xie et al., 2017) explore multi-modal fusion in the same representation space to measure the triple plausibility from multi-views. Advanced multi-modal fusion techniques such as optimal transport (Cao et al., 2022), modality ensemble (Li et al., 2023), contrastive learning (Liang et al., 2024c), and adversarial training (Zhang et al., 2024a) have been continuously introduced into MMKGC. Due to the page limit, a detailed introduction of the existing MMKGC methods can be found in Appendix A.

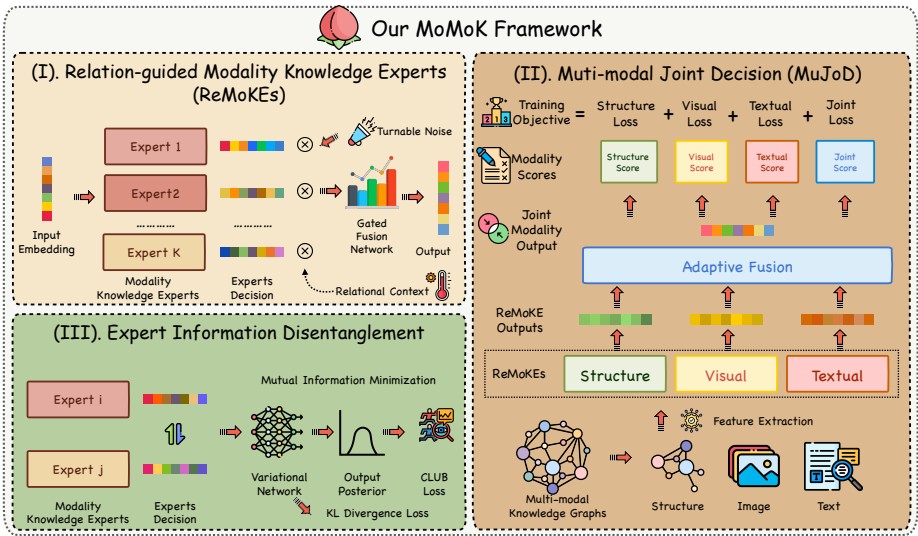

Figure 2: Overview of our proposed MOMOK framework, which consists of three core components: the relation-guided modality knowledge experts (ReMoKE), multi-modal joint decision (MuJoD), and expert information disentanglement (ExID).

**Mixture-of-Experts (MoE)**   MoE is a special model ensemble and combination method that is widely used in AI-related fields like computer vision (Chen et al., 2023b; Riquelme et al., 2021), natural language processing (Jiang et al., 2023; Zhao et al., 2024; 2025), recommendation (Ma et al., 2018; Bian et al., 2023; Hou et al., 2022), and so on. MoE usually divides a given task into multiple subtasks to solve them with individual expert models and design a routing module to select suitable experts to solve the current task. The MoE architecture creates a buzz due to its successful use in large language models (LLMs) (Jiang et al., 2023) as it can efficiently train larger and stronger LLMs.

## 3   PROBLEM DEFINITION

A general KG can be formalized as $\mathcal{KG} = (\mathcal{E}, \mathcal{R}, \mathcal{T})$ where $\mathcal{E}, \mathcal{R}$ are the entity set, the relation set respectively. $\mathcal{T} = \{(h, r, t) \mid h, t \in \mathcal{E}, r \in \mathcal{R}\}$ is the triple set. Furthermore, MMKGs have a modality set denoted as $\mathcal{M}$, encapsulating different modalities in the MMKGs. For an entity $e \in \mathcal{E}$, its modality information of modality $m \in \mathcal{M}$ is denoted as $\mathcal{X}_m(e)$. For different modalities, the elements in it have different forms. For instance, $\mathcal{X}_m(e)$ can be a set of images for image modality and some video clips for video modality. Note that the triple structure $(S)$ is also an extra modality where the structural information is embodied in the triple set $\mathcal{T}$.

MMKGC designs a score function $\mathcal{S}(h, r, t) : \mathcal{E} \times \mathcal{R} \times \mathcal{E} \to \mathbb{R}$ to discriminate the plausibility of a given triple $(h, r, t)$. In this context, a higher score implies a more plausible triple. Entities and relations are embedded into continuous vector spaces for data-driven learning. For MMKGs, all the modality information of each entity will be represented as modality embeddings $\boldsymbol{e}_m (m \in \mathcal{M})$ to participate in the triple score calculation by multi-modal fusion and integration. During training, negative sampling (NS) is widely used to construct manual negative triples for contrastive learning because KGs only have observed positive triples. The negative triples are denoted as:

$$\mathcal{T}' = \{(h', r, t) \mid (h, r, t) \in \mathcal{T} \cap h' \in \mathcal{E} - \{h\}\} \cup \{(h, r, t') \mid (h, r, t) \in \mathcal{T} \cap t' \in \mathcal{E} - \{t\}\} \quad (1)$$

which is generated by a random replacement of entities in the positive triple. During inference, the MMKGC model is usually evaluated with the link prediction task (Bordes et al., 2013) to predict the missing head or tail entity in the given query $(?, r, t)$ or $(h, r, ?)$. For each candidate $e \in \mathcal{E}$, the score of the triple $(h, r, e)$ or $(e, r, t)$ is calculated and then ranked across the entire candidate set.

## 4   METHODOLOGY

In this section, we will present our proposed framework called **M**ix **of** **Mo**dality **K**nowledge experts (MOMOK for short) to achieve robust MMKGC. There are three key components to our design:

relation-guided modality knowledge experts (ReMoKE for short), multi-modal joint decision (MuJoD for short), and expert information disentanglement (ExID for short).

## 4.1 RELATION-GUIDED MODALITY KNOWLEDGE EXPERTS

To better learn the embedding of different perspectives intra-modalities, we introduce a module called relation-guided modality knowledge experts (ReMoKE) to build expert networks in each modality. First, for each modality $m \in \mathcal{M}$, the entity $e \in \mathcal{E}$ possesses a raw modality feature $e_m$, derived from the modality data $\mathcal{X}_m(e)$. For image and text modality, a pre-trained model like VGG (Simonyan & Zisserman, 2015) and BERT (Devlin et al., 2019) would be employed to extract the raw modality feature. As for the structure modality, the raw modality feature will be learned from scratch with the triple data during training.

We then learn the multi-pespective embeddings $\mathcal{V}^e_{m,1}, \mathcal{V}^e_{m,2}, \cdots, \mathcal{V}^e_{m,K}$ of the entity $e$ and modality $m$ by establishing $K$ modality knowledge experts (MoKE) for each modality denoted as $\mathcal{W}_{m,1}, \mathcal{W}_{m,2}, \cdots, \mathcal{W}_{m,K}$. This process can be represented as $\mathcal{V}^e_{m,i} = \mathcal{W}_{m,i}(e_m)$. Then we design a relation-guided gated fusion network (GFN) to facilitate **intra-modality entity embedding fusion** with relation guidance. The output entity embedding for modality $m$ and relation $r$ is denoted as $\widehat{e}_{m,r} = \sum_{i=1}^{K} G_i(\mathcal{V}^e_{m,i}, r)\mathcal{V}^e_{m,i}$ where $G_i$ is the weight for each MoKE calculated by the GFN:

$$G_i(\mathcal{V}^e_{m,i}, r) = \frac{\exp\left((\mathcal{U}_m(\mathcal{V}^e_{m,i}) + \delta_{m,i})/\sigma(\varepsilon_r)\right)}{\sum_{j=1}^{K} \exp\left((\mathcal{U}_m(\mathcal{V}^e_{m,j}) + \delta_{m,j})/\sigma(\varepsilon_r)\right)}, \quad \text{where} \quad \delta_{m,i} \sim \mathcal{N}(0, \mathcal{U}'_m(\mathcal{V}^e_{m,i})) \quad (2)$$

where $\mathcal{U}_m, \mathcal{U}'_m$ are two projection layers and $\delta_{m,i}$ is tunable Gaussian (Shazeer et al., 2017) noise to balance the weights for each MoKE and augment the robustness of the MMKGC model. This is a design (Shazeer et al., 2017) that has been proven to work. Besides, we add a learnable relation-aware temperature $\varepsilon_r$ with a sigmoid function $\sigma$ to limit the temperature in the range $(0, 1)$. We aim to acquire an entity modality embedding within the relational context of the current prediction before making the final decision. This approach allows us to introduce the relational context to guide the modality embedding learning in MoKEs, thereby enabling our MoKEs to extract relation-aware modality embeddings. Besides, the MoKEs will be differentiated to adapt to different relational contexts with the design of GFN. We can learn dynamical modality embeddings of entities that change in different relational contexts.

## 4.2 MULTI-MODAL JOINT DECISION

With the ReMoKE module, we can obtain relation-guided modality embeddings $\widehat{e}_{m,r}(m \in \mathcal{M})$ for each entity under relational context. Subsequently, we equip the model with the ability to amalgamate information from various modalities to facilitate joint decision-making via MuJoD module. MuJoD first accomplishes multi-modal entity embedding fusion by learning a group of adaptive weights for each entity as:

$$\widehat{e}_{Joint,r} = \frac{\exp(\mathcal{W}_{attn} \odot \mathcal{P}_m(\widehat{e}_{m,r}))}{\sum_{n \in \mathcal{M}} \exp(\mathcal{W}_{attn} \odot \mathcal{P}_n(\widehat{e}_{n,r}))} \mathcal{P}_m(\widehat{e}_{m,r}) \quad (3)$$

where $\mathcal{P}_m(m \in \mathcal{M})$ is a projection layer for modality transformation, $\mathcal{W}_{attn}$ is a learnable attention vector shared by each modality, and $\odot$ is the product operator. The joint embedding $\widehat{e}_{Joint}$ aggregates information from all modalities and we treat it as another new "modality" $J$ (short for joint).

We further employ Tucker (Balazevic et al., 2019) score function $\mathcal{S}_m(m \in \mathcal{M})$ to measure the triple plausibility from each modality's perspective, which is denoted as:

$$\mathcal{S}_m(h, r, t) = \mathcal{W}_m \times_1 \widehat{h}_{m,r} \times_2 r_m \times_3 \widehat{t}_{m,r} \quad (4)$$

where $\times_i$ represents the tensor product along the i-th mode, $r_m$ is the learnable embedding of relation $r$ for each modality, $\mathcal{W}_m$ is the core tensor learned during training. We train our model with cross-entropy loss for each triple. For a given triple $(h, r, t)$, we treat $t$ as the golden label for tail prediction against the whole entity set $\mathcal{E}$ and $h$ as the golden label for head prediction, which is the negative sampling process mentioned before. The training objective of each modality $m \in \mathcal{M} \cup \{J\}$ can be denoted as:

$$\mathcal{L}_m = - \sum_{(h,r,t) \in \mathcal{T}} \left( \log \frac{\exp(\mathcal{S}_m(h, r, t))}{\sum_{h' \in \mathcal{E}} \exp(\mathcal{S}_m(h', r, t))} + \log \frac{\exp(\mathcal{S}_m(h, r, t))}{\sum_{t' \in \mathcal{E}} \exp(\mathcal{S}_m(h, r, t'))} \right) \quad (5)$$

This is the standard KGC model training objective, which MuJoD extends to train a separate scoring function for each modality. The overall training objective of MuJoD can be denoted as:

$$\mathcal{L}_{kgc} = \sum_{m \in \mathcal{M} \cup \{J\}} \mathcal{L}_m \tag{6}$$

Since these objectives from different modalities have consistent prediction targets, we directly combine them to derive the final loss $\mathcal{L}_{kgc}$. In the design of MoMoK, we construct intra-modality experts to learn relation-guided embeddings in ReMoKE, and further collectively combine these inter-modality decisions to make more thoughtful predictions. Each modality serves as a senior expert, making decisions in collaboration with the insights from the intra-modality junior experts (single networks in ReMoKE). This hierarchical expert network architecture enables the progressive delivery of valuable entity modal information.

### 4.3 EXPERT INFORMATION DISENTANGLEMENT

Additionally, to further allow the model to learn multi-perspective embeddings guided by the relational context, we propose another expert information disentanglement (ExID) module to disentangle the experts' decisions in each modality based on contrastive log-ratio upper bound (CLUB) (Cheng et al., 2020), which minimizes the mutual information between the multi-perspective embeddings for each modality using CLUB. The general formulation of CLUB can be denoted as:

$$\mathrm{I}_{\mathrm{CLUB}}(\boldsymbol{x}; \boldsymbol{y}) := \mathbb{E}_{p(\boldsymbol{x},\boldsymbol{y})}[\log p(\boldsymbol{y} \mid \boldsymbol{x})] - \mathbb{E}_{p(\boldsymbol{x})}\mathbb{E}_{p(\boldsymbol{y})}[\log p(\boldsymbol{y} \mid \boldsymbol{x})] \tag{7}$$

where $\boldsymbol{x}; \boldsymbol{y}$ are two random variables. $\mathrm{I}_{\mathrm{CLUB}}(\boldsymbol{x}; \boldsymbol{y})$ is an upper bound estimation of their mutual information $\mathrm{I}(\boldsymbol{x}; \boldsymbol{y})$. By optimizing such an objective, we can achieve information disentangling between $\boldsymbol{x}$ and $\boldsymbol{y}$. Since the true conditional probability distribution $p(\boldsymbol{y} \mid \boldsymbol{x})$ is difficult to observe, we can use a variational approximation network to approximate this probability distribution. For the above analysis, we provide more detailed formula derivations and proofs in the **Appendix B**. In our implementation, for modality $m$ with $K$ MoKEs, we disentangle the multi-perspective embeddings $\mathcal{V}_{m,i}^e (1 \leq i \leq K)$ of K MoKEs from each other by the following CLUB mutual information estimation:

$$\mathcal{L}_{club} = \frac{1}{K^2} \sum_{m \in \mathcal{M}} \sum_{e \in \mathcal{B}} \sum_{i=1}^{K} \sum_{j \neq i}^{K} \left( \log \mathcal{Q}_{\theta,m}(\mathcal{V}_{m,j}^e | \mathcal{V}_{m,i}^e) - \sum_{e' \in \mathcal{B} - \{e\}} \log \mathcal{Q}_{\theta,m}(\mathcal{V}_{m,j}^{e'} | \mathcal{V}_{m,i}^e) \right) \tag{8}$$

where $\mathcal{B}$ is a batch of entities and $\mathcal{Q}_{\theta,m}(y|x)$ is the variational approximation of ground-truth posterior of $y$ given $x$ parameterized by a neural network $\theta$ for modality $m$. $e'$ is another entity sampled from the batch $\mathcal{B}$. With such contrastive loss, we can then make MoKEs minimize the mutual information between decisions. Meanwhile, $\mathcal{Q}_{\theta,m}$ should also be trained to o minimize the KL-divergence between the real conditional probabilities distribution $P(\mathcal{V}_{m,j}^e | \mathcal{V}_{m,i}^e)$ and the variational approximation $\mathcal{Q}_{\theta,m}(\mathcal{V}_{m,j}^e | \mathcal{V}_{m,i}^e)$ by optimize the Kullback-Leibler divergence:

$$\mathcal{L}_{exid} = \mathbb{D}_{KL} \left[ P(\mathcal{V}_{m,j}^e | \mathcal{V}_{m,i}^e) || \mathcal{Q}_{\theta,m}(\mathcal{V}_{m,j}^e | \mathcal{V}_{m,i}^e) \right] \tag{9}$$

which will be alternatively optimized with the main MMKGC model during training. Here, the real conditional distribution is usually assumed as a Gaussian distribution (Cheng et al., 2020).

### 4.4 TRAINING AND INFERENCE

Combining all the designs above, the final objective for our MMKGC model can be represented as:

$$\mathcal{L} = \mathcal{L}_{kgc} + \lambda \mathcal{L}_{club} \tag{10}$$

We collectively train the embeddings with prediction losses $\mathcal{L}_m$ from each modality in a multi-task manner. The disentangle loss $\mathcal{L}_{club}$ is regulated by a weight $\lambda$. Besides, during each round of training, $\mathcal{Q}_{\theta,m}$ is also optimized with the loss $\mathcal{L}_{exid}$, separated from the MMKGC model. During the inference stage, we calculate the joint score for each triple as $\mathcal{S}(h, r, t) = \sum_{m \in \mathcal{M} \cup \{J\}} \mathcal{S}_m(h, r, t)$ which considers the contribution from each modality and provides a full-view prediction. This score function $\mathcal{S}(h, r, t)$ will be the final measurement of the triple plausibility and applied for candidate triple ranking and evaluation.

# 5 EXPERIMENTS AND EVALUATION

In this section, we will introduce the basic experiment settings of our work and demonstrate our evaluation results with extensive analysis. The following five research questions (RQ) are the key questions that we explore in the experiments.

**RQ1.** Can MOMOK outperform the existing baselines and achieve state-of-the-art performance?

**RQ2.** Can MOMOK maintain robust performance tasks when the modality information is noisy?

**RQ3.** How much do each module in the MOMOK contribute to the final performance?

**RQ4.** How does the training efficiency of our model compare to existing methods?

**RQ5.** Are there intuitive cases to straightly demonstrate the effectiveness of MOMOK?

## 5.1 DATASETS

We conduct our experiments on four public MMKG benchmarks: MKG-W (Xu et al., 2022), MKG-Y (Xu et al., 2022), DB15K (Liu et al., 2019), and KVC16K (Zhang et al., 2024a). MKG-W and MKG-Y are the subsets of Wikidata (Vrandecic & Krötzsch, 2014), YAGO (Suchanek et al., 2007), and DBPedia (Lehmann et al., 2015) respectively. KVC16K is modified from KuaiPedia (Pan et al., 2022), a micro-video encyclopedia. They are all real-world MMKGs with image and text modalities. The detailed information on the datasets can be found in Table 5 in Appendix.

## 5.2 EXPERIMENTAL SETTINGS

**Baseline Methods** To make comprehensive comparisons, we chose 20 recent SOTA MMKGC methods as the baselines for the experiments. The first category is uni-modal KGC methods including TransE (Bordes et al., 2013), DistMult (Yang et al., 2015), ComplEx (Trouillon et al., 2016), RotatE (Sun et al., 2019), PairRE (Chao et al., 2021), and TuckER (Balazevic et al., 2019). The second category is multi-modal KGC methods considering multi-modal information of entities to enhance the KGC models, including IKRL (Xie et al., 2017), TBKGC (Sergieh et al., 2018), TransAE (Wang et al., 2019b), RSME (Wang et al., 2021), MMKRL (Lu et al., 2022), VBKGC (Zhang & Zhang, 2022), OTKGE (Cao et al., 2022), MoSE (Zhao et al., 2022), MMRNS (Xu et al., 2022), MANS (Zhang et al., 2023a), IMF (Li et al., 2023), QEB (Wang et al., 2023), VISTA (Lee et al., 2023), and AdaMF (Zhang et al., 2024b). Among these methods, MoSE and IMF are two methods using ensemble learning technologies, which have similarities to our design and are worth making comparisons. The third category is negative sampling methods including MANS (Zhang et al., 2023a) and MMRNS (Xu et al., 2022). Please refer to Appendix C.1 for more detailed information.

**Task and Evaluation Protocols** We evaluate the MMKGC models with the link prediction task (Bordes et al., 2013), the most popular KGC task. We use rank-based metrics like mean reciprocal rank (MRR) (Sun et al., 2019), and Hit@K (K=1, 3, 10)(Bordes et al., 2013) to quantitatively evaluate the link prediction performance, considering both head prediction $(h, r, ?)$ and tail prediction $(?r, t)$. Filter setting (Bordes et al., 2013) is used to eliminate the effect of triples that have already appeared in the training data. Please refer to Appendix C.2 for more detailed information.

**Implementation Details** In our experiments, we implement our method with PyTorch and conduct each experiment on a Linux server with the Ubuntu 20.04.1 operating system and a single NVIDIA A800 GPU. The variational approximation network $\theta$ and the projection layers are all implemented by two-layer MLPs with ReLU as activation (Glorot et al., 2011). During training, we set the batch size to 1024. The embedding dimension $d$ is tuned from $\{200, 250, 300, 400, 500\}$. We optimize the model with Adam (Kingma & Ba, 2015) and the learning rate is tuned from $\{1e^{-3}, 5e^{-4}, 1e^{-4}\}$. The loss weight $\lambda$ is tuned in $\{1e^{-3}, 1e^{-4}, 1e^{-5}\}$. The number K of experts in ReMoKE is set to 3. Each experiment takes 1-3 hours to accomplish across different datasets. For baselines, we reproduce the results following the settings described in the original papers and their open-source official code. Some of the baseline results refer to MMRNS (Xu et al., 2022).

Table 1: The main MMKGC results on four datasets. The best results are bold and the second best results are underlined. Methods with special mark * are ensemble-based methods.

| Model | MKG-W | | MKG-Y | | DB15K | | | | KVC16K | | | |
|---|---|---|---|---|---|---|---|---|---|---|---|---|
| | MRR | Hit@1 | MRR | Hit@1 | MRR | Hit@1 | Hit@3 | Hit@10 | MRR | Hit@1 | Hit@3 | Hit@10 |
| *Uni-modal KGC Methods* | | | | | | | | | | | | |
| **TransE** | 29.19 | 21.06 | 30.73 | 23.45 | 24.86 | 12.78 | 31.48 | 47.07 | 8.54 | 0.64 | 10.97 | 23.42 |
| **DistMult** | 20.99 | 15.93 | 25.04 | 19.33 | 23.03 | 14.78 | 26.28 | 39.59 | 6.37 | 3.03 | 6.11 | 12.61 |
| **ComplEx** | 24.93 | 19.09 | 28.71 | 22.26 | 27.48 | 18.37 | 31.57 | 45.37 | 12.85 | 7.48 | 13.79 | 23.18 |
| **RotatE** | 33.67 | 26.80 | 34.95 | 29.10 | 29.28 | 17.87 | 36.12 | 49.66 | 14.33 | 8.25 | 15.37 | 26.17 |
| **PairRE** | 34.40 | 28.24 | 32.01 | 25.53 | 31.13 | 21.62 | 35.91 | 49.30 | - | - | - | - |
| **TuckER** | 29.59 | 23.93 | 37.05 | 34.59 | 33.86 | 25.34 | 37.91 | 50.38 | 15.90 | 9.79 | 17.24 | 27.58 |
| *Multi-modal KGC Methods* | | | | | | | | | | | | |
| **IKRL** | 32.36 | 26.11 | 33.22 | 30.37 | 26.82 | 14.09 | 34.93 | 49.09 | 11.11 | 5.42 | 11.46 | 22.39 |
| **TBKGC** | 31.48 | 25.31 | 33.99 | 30.47 | 28.40 | 15.61 | 37.03 | 49.86 | 5.39 | 0.35 | 5.04 | 15.52 |
| **TransAE** | 30.00 | 21.23 | 28.10 | 25.31 | 28.09 | 21.25 | 31.17 | 41.17 | 10.81 | 5.31 | 11.34 | 21.89 |
| **MMKRL** | 30.10 | 22.16 | 36.81 | 31.66 | 26.81 | 13.85 | 35.07 | 49.39 | 8.78 | 3.89 | 8.99 | 18.34 |
| **RSME** | 29.23 | 23.36 | 34.44 | 31.78 | 29.76 | 24.15 | 32.12 | 40.29 | 12.31 | 7.14 | 13.21 | 22.05 |
| **VBKGC** | 30.61 | 24.91 | 37.04 | 33.76 | 30.61 | 19.75 | 37.18 | 49.44 | 14.66 | 8.28 | 15.81 | 27.04 |
| **OTKGE** | 34.36 | 28.85 | 35.51 | 31.97 | 23.86 | 18.45 | 25.89 | 34.23 | 8.77 | 5.01 | 9.31 | 15.55 |
| **MoSE*** | 33.34 | 27.78 | 36.28 | 33.64 | 28.38 | 21.56 | 30.91 | 41.67 | 8.81 | 4.75 | 9.46 | 16.40 |
| **IMF*** | 34.50 | 28.77 | 35.79 | 32.95 | 32.25 | 24.20 | 36.00 | 48.19 | 12.01 | 7.42 | 12.82 | 21.01 |
| **QEB** | 32.38 | 25.47 | 34.37 | 29.49 | 28.18 | 14.82 | 36.67 | 51.55 | 12.06 | 5.57 | 13.03 | 25.01 |
| **VISTA** | 32.91 | 26.12 | 30.45 | 24.87 | 30.42 | 22.49 | 33.56 | 45.94 | 11.89 | 6.97 | 12.66 | 21.27 |
| **AdaMF** | 34.27 | 27.21 | **38.06** | 33.49 | 32.51 | 21.31 | 39.67 | 51.68 | 15.26 | 8.56 | 16.71 | 28.29 |
| *Negative Sampling Methods* | | | | | | | | | | | | |
| **MANS** | 30.88 | 24.89 | 29.03 | 25.25 | 28.82 | 16.87 | 36.58 | 49.26 | 10.42 | 5.21 | 11.01 | 20.45 |
| **MMRNS** | 35.03 | 28.59 | 35.93 | 30.53 | 32.68 | 23.01 | 37.86 | 51.01 | 13.31 | 7.51 | 14.19 | 24.68 |
| **MoMoK** | **35.89** | **30.38** | 37.91 | **35.09** | **39.57** | **32.38** | **43.45** | **54.14** | **16.87** | **10.53** | **18.26** | **29.20** |
| Improvements | +2.5% | +4.2% | - | +3.9% | +21.1% | +33.8% | +9.5% | +4.8% | +10.6% | +23.0% | +9.3% | +3.21% |

## 5.3 MAIN RESULTS (RQ1)

The main MMKGC results are detailed in Table 1. Comparison with the recent 19 baselines reveals that MoMoK makes significant progress in almost all the metrics and achieves new state-of-the-art results. When contrasted with existing ensemble-based approaches such as MoSE (Zhao et al., 2022) and IMF (Li et al., 2023), MoMoK excels by fully exploiting the potential of the relational context. These methods often merely assign weights to models across different modalities, overlooking the impact of intricate factors like relational context. In contrast, MoMoK thoroughly incorporates these considerations.

Furthermore, it is evident that MoMoK achieves most pronounced improvements in Hit@1 across different metrics. For instance, MoMoK obtained 33.8% / 23.0% relative improvement of Hit@1 on DB15K and KVC16K respectively. This underscores that, compared with baseline models, our method is more effective at ranking correct answers first. It demonstrates the significant contribution of our multi-modal information utilization and relational context to the refined, accurate reasoning capabilities of the MMKGC model.

## 5.4 MMKGC EXPERIMENTS IN COMPLEX ENVIRONMENTS (RQ2)

To assess the robustness of our method in complex scenarios, we conducted MMKGC experiments in three different scenarios: modality noisy scenario, modality missing scenario, and link sparse scenario. For the modality noisy scenario, we set a noise ratio for the MMKG dataset, according to which a portion of the solid modal information is added with Gaussian noise before performing the MMKGC experiments. For the modality missing scenario, we randomly remove some of the modality information for a certain proportion of entities. For a link sparse scenario, we randomly remove some of the training triples and create a data-sparse environment.

As presented in Figure 3, the experimental results indicate that MoMoK continues to outperform the baseline method despite these conditions. It is evident that the complex environments significantly affect the MMKGC prediction at coarse grains, as indicated by the more pronounced volatility of Hit@10 compared to MRR. The variation in Hit@10 results reveals that baseline methods like TBKGC (Sergieh et al., 2018) and AdaMF (Zhang et al., 2024b) undergo a noticeable performance degradation with increasing noise and decreasing training data, while our method's performance remains relatively steady. From the above implementation results, our approach achieves better results compared to baseline in a variety of complex scenarios.

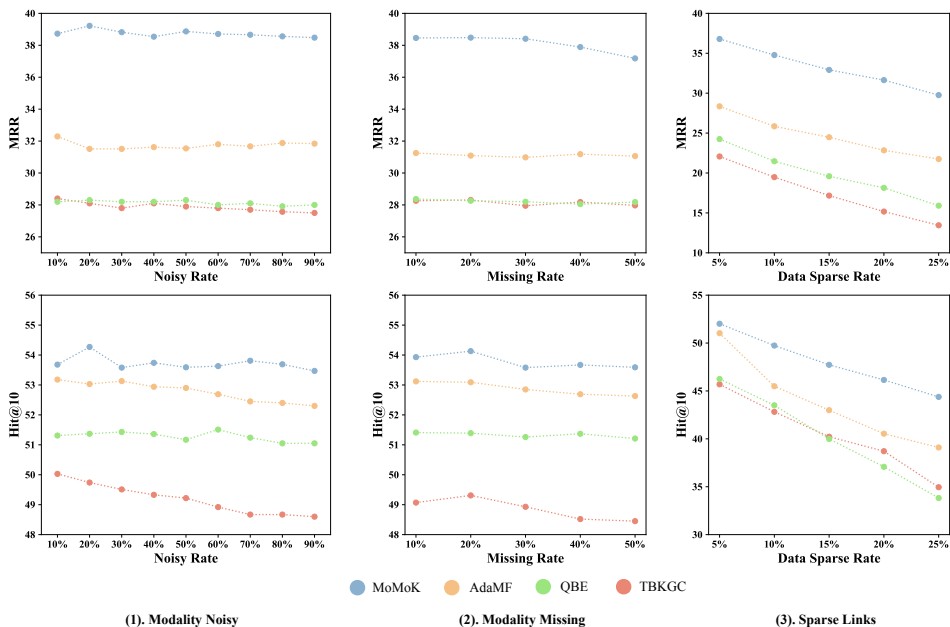

**(1). Modality Noisy**  **(2). Modality Missing**  **(3). Sparse Links**

Figure 3: MMKGC results (MRR and Hit@10) of DB15K dataset under three different scenario: modality noisy, modality missing, and link sparse. We compare our method MoMoK with three recent MMKGC baselines AdaMF, TBKGC, and QBE.

Table 2: The ablation study results on MKG-W and DB15K datasets. We explored the impact of the design of each modality already each important component on the final result.

| Setting | | MKG-W | | DB15K | | | |
|---|---|---|---|---|---|---|---|
| | | MRR | Hit@1 | MRR | Hit@1 | Hit@3 | Hit@10 |
| Full Model | | 35.89 | 30.38 | 39.57 | 32.38 | 43.45 | 54.14 |
| Modality Contribution | (1.1). Structure Modality | 32.82 | 27.73 | 36.45 | 29.36 | 39.99 | 49.86 |
| | (1.2). Image Modality | 32.75 | 27.78 | 36.84 | 29.80 | 40.10 | 50.42 |
| | (1.3). Text Modality | 32.62 | 27.66 | 37.04 | 29.93 | 40.49 | 50.39 |
| | (1.4). Joint Modality | 34.76 | 29.33 | 36.87 | 29.90 | 42.44 | 53.93 |
| Model Design | (2.1). w/o relational $\epsilon_r$ | 35.50 | 29.98 | 39.40 | 31.47 | 43.19 | 52.88 |
| | (2.2). w/o noise $\delta_m$ | 35.31 | 29.69 | 39.43 | 31.54 | 43.32 | 53.75 |
| | (2.3). w/o adaptive fusion | 35.34 | 30.04 | 39.01 | 30.74 | 43.29 | 53.85 |
| | (2.4). w/o joint training | 32.73 | 27.09 | 37.62 | 29.72 | 41.64 | 52.73 |
| | (2.5). w/o ExID | 34.99 | 29.49 | 38.42 | 30.63 | 42.42 | 53.24 |

## 5.5 ABLATION STUDY (RQ3)

To confirm the soundness of our design, we conduct further ablation studies to investigate the contribution of each module in MoMoK. Our ablation experiments are divided into two main parts. The first part aims to analyze the information from each modality and validate whether they positively contribute to the performance. The second part is dedicated to examining our designs in MoMoK (ReMoKE, MuJoD, ExID) and verifying whether their design has rationality by removing the corresponding modules. The experimental results are presented in Table 2.

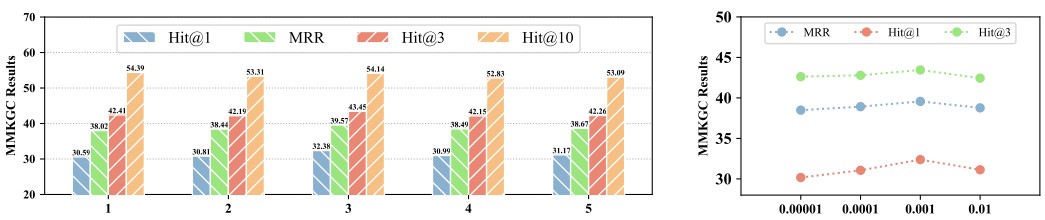

Figure 4: Additional parameter analysis about the number of MoKEs and the weight $\lambda$ for the $\mathcal{L}_{club}$.

Table 4: A case study of MOMOK. We list some of the relations that are predicted best by each modality score $\mathcal{S}_m$ to verify the contribution of each modality to the final result.

| Modality | Outperforming Predictions |
|---|---|
| Structure | RouteJunction, PartOf, Nearest City, Publisher, PrimeMinister, ComputingPlatform, LargestCity |
| Image | SisterStation, League, Parent, HubAirport, Company, Owner, Capital |
| Text | SisterStation, Publisher, Head, FederalState, Parent, ComputingPlatform, CountrySeat |
| Joint | GoverningBody, PartOf, Creator, Company, ComputingPlatform, RegionServed |
| Full Model | SisterStation, RouteJunction, GoverningBody, Publisher, PartOf, FederalState, ComputingPlatform, Parent |

From the first group of experimental results we can observe that each modality's information contributes to the final result, and during training we set up a separate model for each modality, with their respective performance on two datasets being lower than the full model result.

Moreover, the results from the second group reveal that some of our key designs in the three modules significantly contribute to the final performance. Experiments (2.1) and (2.2) confirm the effectiveness of relational context and tunable noise in the ReMoKE module. Experiments (2.3) and (2.4) focus on the MuJoD module and the results proved the effectiveness of the adaptive fusion (Equation 3) and joint training (Equation 6). Experiment (2.5) further examines the impact of the CLUB loss on information disentanglement. Collectively, these findings indicate that joint training has the most profound effect on the final performance, as it trains a separate MMKGC model for each modality, resulting in decision fusion.

We also investigate the effect of some crucial hyperparameters, such as the number of experts $K$ in the ReMoKE module, and the weights of the ExID loss $\lambda$, as depicted in Figure 4. It can be observed that the impact of the number of experts K on the final results generally follows a pattern of initial increase followed by a decrease, mainly affecting fine-grained metrics such as Hit@1 and MRR. Having either too many or too few experts is detrimental to the model's learning performance. The impact of weight $\lambda$ is similar. The model achieves the best results at $K = 3$ and $\lambda = 0.0001$.

## 5.6 EFFICIENCY ANALYSIS (RQ4)

Even though our design introduces many new modules that have not appeared in existing methods, there is not much loss of efficiency in terms of time and space complexity. The introduction of new computational and parametric quantities occurs in two main sections, the MuJoD module and the ExID module. Our approach remains linear in the growth of complexity, and there is no such level of latency as an exponential explosion. Here, we present the time efficiency and GPU memory usage in Table 3. Combining the SOTA MMKGC performance of MOMOK in Table 1 and the training ef-

| Method | Time | GPU Memory |
|---|---|---|
| MoMoK | 9.8s | 5900MB |
| MMKRL | 7.5s | 4504MB |
| OTKGE | 70.1s | 2540MB |
| MMRNS | 25.5s | 25582MB |

Table 3: Time and GPU memory cost for different MMKGC methods.

ficiency/GPU consumption results presented in this table we can conclude that our approach achieves high efficiency and less GPU memory usage while maintaining the SOTA effect. Therefore, the time and space complexity of the present method is within reasonable limits and still performs well.

## 5.7 CASE STUDY (RQ5)

To provide a more intuitive justification and interpretability for our approach, we conduct the case study from both macroscopic and microscopic viewpoints. We set a separate score for each modality in the MuJoD module and finally integrate them for joint decision-making. Therefore, to visualize the contribution of each modality to the final result, we list in Table 4 several relations where each modality score achieves the best results.

Notably, the relation types that each modality score $\mathcal{S}_m$ best predicts are diverse. These relations that perform best in the overall prediction can be found in the prediction results of the different modalities. This implies that MOMOK effectively merges predictions from various modalities for joint consideration, thereby outperforming the results achieved by individual modalities.

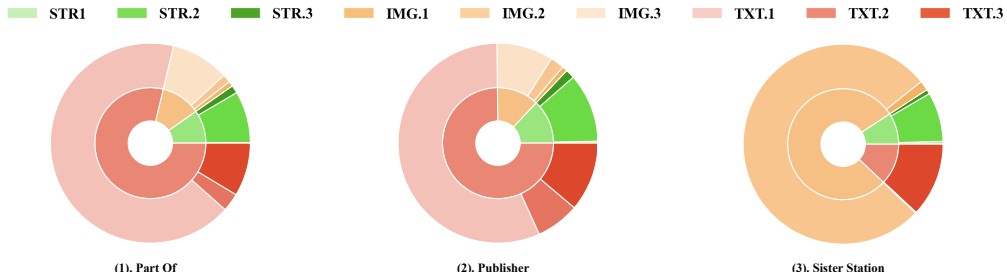

Figure 5: Attention weights visualization results. We select some relations and present the weights of each modality contributing to the joint representation $\widehat{e}_{Joint}$. We further present the weights $G_i$ for $K(K=3)$ ReMoKEs in the modality outputs $\widehat{e}_m$. Abbreviations for modalities: Structure (STR), Image (IMG), Text (TXT). M.k in the legend denotes the k-th expert of modality M.

Simultaneously, we delve into the micro level by analyzing the adaptive weights in MoMoK. Our design incorporates the expert decisions via a series of adaptive weights in the ReMoKE, while the MuJoD module also employs adaptive weights to derive the joint modality embedding from the outputs of the modalities. We select a handful of relations to investigate the weights from each modality and each ReMoKE within the joint modality embedding of the entities in the respective relational contexts.

As shown in Figure 5, the joint embedding of entities in varied relational contexts assigns diverse significance to each modality's information. For example, *PartOf* attaches more weight to textual modality, while *Parents* relies more on image modality. Furthermore, the majority of contributions within each modality come from the same expert, and the contributions from different modalities in distinct relational contexts vary greatly. This indicates that we successfully delegate different ReMoKEs intra-modality to handle different relational contexts, aligning with our original intent of proposing the MoE architecture to address the MMKGC task.

## 6    CONCLUSION AND FUTURE WORK

In this paper, we propose a new MMKGC framework called MoMoK to learn modality features in diverse perspectives from the raw modality information of entities with relational guidance and integrate the multi-modal information through modality knowledge experts. We further decouple the expert networks and enhance the model's expressive capability through the comparative estimation of mutual information. Experimental results show that our design can achieve new SOTA results on multiple public benchmarks with both robustness, reasonability, and interpretability. Looking ahead, we can further design a more rational MoE architecture that not only accomplishes the tasks of the MMKGC but also finds ways to incorporate the MMKG and the large language models to realize a sparse large language model with multi-modal knowledge perception.

### ACKNOWLEDGMENTS

This work is founded by National Natural Science Foundation of China (NSFCU23B2055 / NS-FCU19B2027 / NSFC62306276), Zhejiang Provincial Natural Science Foundation of China (No. LQ23F020017), Yongjiang Talent Introduction Programme (2022A-238-G), and Fundamental Research Funds for the Central Universities (226-2023-00138). This work was supported by AntGroup.

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

## A    RELATED WORKS

Multi-modal Knowledge Graph Completion (MMKGC) aims to achieve missing knowledge discovery in the given knowledge graph by considering both structural information and the multi-modal content of entities. Existing MMKGC methods can be divided into three categories based on the design ideas:

- **Modality Fusion Methods**. These methods integrate multi-modal embeddings of entities with their structural embeddings for triple plausibility estimation. Early approaches like IKRL (Xie et al., 2017), TBKGC (Sergieh et al., 2018), TransAE (Wang et al., 2019b) employ multiple translation-based score functions for embeddings from diverse modalities. Later approaches such as OTKGE (Cao et al., 2022), VISTA (Lee et al., 2023), and AdaMF (Zhang et al., 2024b) introduced more sophisticated feature fusion techniques such as optimal transfer, transformer, and adversarial training into multi-modal fusion.
- **Modality Ensemble Methods**. These methods train separate models for each modality and combine them for a joint final decision. MoSE (Zhao et al., 2022) design three different ensemble methods to adaptively combine the single-modality models. IMF (Li et al., 2023) designs an interactive module among modalities with both fusion and ensemble.
- **Modality-aware Negative Sampling Methods**. These methods involve the negative sampling design by considering the extra multi-modal information for better contrastive learning during the MMKGC model training process. For example, MANS (Zhang et al., 2023a) designs visual negative sampling strategies to align the visual modality with a triple structure. MMRNS (Xu et al., 2022) introduces a relation-enhanced negative sampling method by considering relation-guided negative sample generation.

In this paper, our research focuses on designing a mixture-of-experts framework with ensemble learning for MMKGC, which would be the combination of the modality fusion and modality ensemble methods.

## B    MODEL DESIGN AND ANALYSIS

In this section, we present a detailed theoretical analysis of the CLUB loss Cheng et al. (2020) we used in Section 4.3.

**Theorem 1**   : For two entity embedding $\mathcal{V}_{m,i}^e, \mathcal{V}_{m,j}^e$ (simplified as $\mathcal{V}_i$ and $\mathcal{V}_j$ in the following derivation for the sake of typography), they can be treated as two random variables. The upper bound of their mutual information $\mathrm{I}(\mathcal{V}_i; \mathcal{V}_j)$ is:

$$\mathrm{I}_{\mathrm{CLUB}}(\mathcal{V}_i; \mathcal{V}_j) := \mathbb{E}_{p(\mathcal{V}_i,\mathcal{V}_j)}[\log p(\mathcal{V}_j \mid \mathcal{V}_i)] - \mathbb{E}_{p(\mathcal{V}_i)}\mathbb{E}_{p(\mathcal{V}_j)}[\log p(\mathcal{V}_j \mid \mathcal{V}_i)] \quad (11)$$

**Proof 1**   This theorem can be proven with the following derivation. We calculate the difference between $\mathrm{I}_{\mathrm{CLUB}}(\mathcal{V}_i; \mathcal{V}_j)$ and $\mathrm{I}(\mathcal{V}_i; \mathcal{V}_j)$ by:

$$\begin{aligned}
\Delta := \ & \mathrm{I}_{\mathrm{CLUB}}(\mathcal{V}_i; \mathcal{V}_j) - \mathrm{I}(\mathcal{V}_i; \mathcal{V}_j) \\
= \ & \mathbb{E}_{p(\mathcal{V}_i,\mathcal{V}_j)}[\log p(\mathcal{V}_j \mid \mathcal{V}_i)] - \mathbb{E}_{p(\mathcal{V}_i)}\mathbb{E}_{p(\mathcal{V}_j)}[\log p(\mathcal{V}_j \mid \mathcal{V}_i)] - \mathbb{E}_{p(\mathcal{V}_i,\mathcal{V}_j)}[\log p(\mathcal{V}_j \mid \mathcal{V}_i) - \log p(\mathcal{V}_j)] \\
= \ & \mathbb{E}_{p(\mathcal{V}_i,\mathcal{V}_j)}[\log p(\mathcal{V}_j)] - \mathbb{E}_{p(\mathcal{V}_i)}\mathbb{E}_{p(\mathcal{V}_j)}[\log p(\mathcal{V}_j \mid \mathcal{V}_i)] \\
= \ & \mathbb{E}_{p(\mathcal{V}_j)}[\log p(\mathcal{V}_j) - \mathbb{E}_{p(\mathcal{V}_i)}[\log p(\mathcal{V}_j \mid \mathcal{V}_i)]]
\end{aligned}$$

$$(12)$$

Besides, according to Jensen inequality, we have:

$$\begin{aligned}
\log p(\mathcal{V}_j) = \log(\mathbb{E}_{p(\mathcal{V}_i)}[p(\mathcal{V}_j \mid \mathcal{V}_i)]) &\geq \mathbb{E}_{p(\mathcal{V}_i)}[\log p(\mathcal{V}_j \mid \mathcal{V}_i)] \\
\Rightarrow \Delta := \mathrm{I}_{\mathrm{CLUB}}(\mathcal{V}_i; \mathcal{V}_j) - \mathrm{I}(\mathcal{V}_i; \mathcal{V}_j) &\geq 0
\end{aligned} \quad (13)$$

Therefore, $\mathrm{I}_{\mathrm{CLUB}}(\mathcal{V}_i; \mathcal{V}_j)$ is an upper bound of $\mathrm{I}(\mathcal{V}_i; \mathcal{V}_j)$. Besides, the real conditional distribution $p(\mathcal{V}_j \mid \mathcal{V}_i)$ is hard to measure. Therefore, a variational distribution $q_\theta(\mathcal{V}_j \mid \mathcal{V}_i)$ parameterized by a neural network $\theta$ is usually employed to approximate the real distribution. Therefore, the variational CLUB objective can be denoted as:

$$\mathrm{I}_{\mathrm{CLUB}}'(\mathcal{V}_i; \mathcal{V}_j) := \mathbb{E}_{q_\theta(\mathcal{V}_i,\mathcal{V}_j)}[\log q_\theta(\mathcal{V}_j \mid \mathcal{V}_i)] - \mathbb{E}_{q_\theta(\mathcal{V}_i)}\mathbb{E}_{q_\theta(\mathcal{V}_j)}[\log q_\theta(\mathcal{V}_j \mid \mathcal{V}_i)] \quad (14)$$

It can be proved that with a good variational network $q_\theta$, I' can still hold an upper bound of the mutual information. Here we denote $q_\theta(\mathcal{V}_i, \mathcal{V}_j) = q_\theta(\mathcal{V}_j \mid \mathcal{V}_i)p(\mathcal{V}_i)$ and introduce such a new theorem:

**Theorem 2** If we have
$$KL(p(\mathcal{V}_i, \mathcal{V}_j)\|q_\theta(\mathcal{V}_i, \mathcal{V}_j)) \leq KL(p(\mathcal{V}_i)p(\mathcal{V}_j)\|q_\theta(\mathcal{V}_i, \mathcal{V}_j)) \tag{15}$$
then we have $\mathrm{I}(\mathcal{V}_i; \mathcal{V}_j) \leq \mathrm{I}'_{\mathrm{CLUB}}(\mathcal{V}_i; \mathcal{V}_j)$. KL represents the Kullback-Leibler divergence between two probability distributions.

**Proof 2** We can have the following derivation:
$$
\begin{aligned}
\tilde{\Delta} := {}& \mathrm{I}'_{\mathrm{CLUB}}(\mathcal{V}_i; \mathcal{V}_j) - \mathrm{I}(\mathcal{V}_i; \mathcal{V}_j) \\
= {}& \mathbb{E}_{p(\mathcal{V}_i,\mathcal{V}_j)}[\log q_\theta(\mathcal{V}_j \mid \mathcal{V}_i)] - \mathbb{E}_{p(\mathcal{V}_i)}\mathbb{E}_{p(\mathcal{V}_j)}[\log q_\theta(\mathcal{V}_j \mid \mathcal{V}_i)] - \mathbb{E}_{p(\mathcal{V}_i,\mathcal{V}_j)}[\log p(\mathcal{V}_j \mid \mathcal{V}_i) - \log p(\mathcal{V}_j)] \\
= {}& [\mathbb{E}_{p(\mathcal{V}_j)}[\log p(\mathcal{V}_j)] - \mathbb{E}_{p(\mathcal{V}_i)p(\mathcal{V}_j)}[\log q_\theta(\mathcal{V}_j \mid \mathcal{V}_i)] \\
& - [\mathbb{E}_{p(\mathcal{V}_i,\mathcal{V}_j)}[\log p(\mathcal{V}_j \mid \mathcal{V}_i)] - \mathbb{E}_{p(\mathcal{V}_i,\mathcal{V}_j)}[\log q_\theta(\mathcal{V}_j \mid \mathcal{V}_i)]]. \\
= {}& \mathbb{E}_{p(\mathcal{V}_i)p(\mathcal{V}_j)}[\log \frac{p(\mathcal{V}_j)}{q_\theta(\mathcal{V}_j \mid \mathcal{V}_i)}] - \mathbb{E}_{p(\mathcal{V}_i,\mathcal{V}_j)}[\log \frac{p(\mathcal{V}_j \mid \mathcal{V}_i)}{q_\theta(\mathcal{V}_j \mid \mathcal{V}_i)}] \\
= {}& \mathbb{E}_{p(\mathcal{V}_i)p(\mathcal{V}_j)}[\log \frac{p(\mathcal{V}_i)p(\mathcal{V}_j)}{q_\theta(\mathcal{V}_j \mid \mathcal{V}_i)p(\mathcal{V}_i)}] - \mathbb{E}_{p(\mathcal{V}_i,\mathcal{V}_j)}[\log \frac{p(\mathcal{V}_j \mid \mathcal{V}_i)p(\mathcal{V}_i)}{q_\theta(\mathcal{V}_j \mid \mathcal{V}_i)p(\mathcal{V}_i)}] \\
= {}& \mathrm{KL}(p(\mathcal{V}_i)p(\mathcal{V}_j)\|q_\theta(\mathcal{V}_i, \mathcal{V}_j)) - \mathrm{KL}(p(\mathcal{V}_i, \mathcal{V}_j)\|q_\theta(\mathcal{V}_i, \mathcal{V}_j))
\end{aligned}
\tag{16}
$$
Therefore, $\mathrm{I}'_{\mathrm{CLUB}}(\mathcal{V}_i; \mathcal{V}_j)$ is an upper bound of $\mathrm{I}(\mathcal{V}_i; \mathcal{V}_j)$ if and only if $KL(p(\mathcal{V}_i, \mathcal{V}_j)\|q_\theta(\mathcal{V}_i, \mathcal{V}_j)) \leq KL(p(\mathcal{V}_i)p(\mathcal{V}_j)\|q_\theta(\mathcal{V}_i, \mathcal{V}_j))$. The equality holds when $\mathcal{V}_i, \mathcal{V}_j$ are independent.

This theorem indicates that $\mathrm{I}'_{\mathrm{CLUB}}(\mathcal{V}_i; \mathcal{V}_j)$ can still be an upper bound for $\mathrm{I}(\mathcal{V}_i; \mathcal{V}_j)$ if the variational distribution $q_\theta(\mathcal{V}_i, \mathcal{V}_j)$ is closer to $p(\mathcal{V}_i, \mathcal{V}_j)$ than to $p(\mathcal{V}_i)p(\mathcal{V}_j)$. Hence, we can minimizing $KL(p(\mathcal{V}_i, \mathcal{V}_j)\|q_\theta(\mathcal{V}_i, \mathcal{V}_j))$ to achieve this goal.

In our specific implementations, we employ several neural networks $\mathcal{Q}_{\theta,m}$ to estimate the variational distribution. We further design two loss ($\mathcal{L}_{club}$ in Equation 8 and $\mathcal{L}_{exid}$ in Equation 9) to implement the CLUB module. $\mathcal{L}_{club}$ is designed to estimate the mutual information among different MoKEs and $\mathcal{L}_{exid}$ is designed to optimize the neural networks $\mathcal{Q}_{\theta,m}$. These two losses correspond to Theorem 1 and Theorem 2 derived earlier.

## C  EXPERIMENT DETAILS

### C.1  BASELINES

The KGC baselines we compared in our experiments can be divided into three categories:

(1). **Uni-modal KGC methods:** TransE (Bordes et al., 2013), DistMult (Yang et al., 2015), ComplEx (Trouillon et al., 2016), RotatE (Sun et al., 2019), PairRE (Chao et al., 2021), and TuckER (Balazevic et al., 2019). These methods only consider the triple structural information in their designs and learn structural embeddings for KGC.

(2). **Multi-modal KGC methods:** IKRL (Xie et al., 2017), TBKGC (Sergieh et al., 2018), TransAE (Wang et al., 2019b), RSME (Wang et al., 2021), MMKRL (Lu et al., 2022), VBKGC (Zhang & Zhang, 2022), OTKGE (Cao et al., 2022), MoSE (Zhao et al., 2022), MMRNS (Xu et al., 2022), MANS (Zhang et al., 2023a), IMF (Li et al., 2023), QEB (Wang et al., 2023), VISTA (Lee et al., 2023), and AdaMF (Zhang et al., 2024b). These methods employ multi-modal information such as texts and images to enhance the entity representation learning, achieving better KGC performance.

(3). **Negative sampling methods:** MANS (Zhang et al., 2023a) and MMRNS (Xu et al., 2022). These methods attempt to modify the traditional negative sampling methods with the multi-modal information from entities to make fine-grained contrast during training.

In the experiments, we conduct a comprehensive comparison with the mentioned 20 baselines to demonstrate that MoMoK can learn better entity representations to enhance the MMKGC process. Some methods (Yao et al., 2019) fine-tuning the pre-trained models are orthogonal to our design philosophy and paradigm so we do not compare with them.

Table 5: Statistical information of the four MMKGs in our experiments. The image and text modality features are provided by the original datasets and kept the same for all baselines.

| Dataset | #Entity | #Relation | #Train | #Valid | #Test | Image | | Text | |
|---|---|---|---|---|---|---|---|---|---|
| | | | | | | Num | Dim | Num | Dim |
| **MKG-W** (Xu et al., 2022) | 15000 | 169 | 34196 | 4276 | 4274 | 14463 | 383 | 14123 | 384 |
| **MKG-Y** (Xu et al., 2022) | 15000 | 28 | 21310 | 2665 | 2663 | 14244 | 383 | 12305 | 384 |
| **DB15K** (Liu et al., 2019) | 12842 | 279 | 79222 | 9902 | 9904 | 12818 | 4096 | 9078 | 768 |
| **KVC16K** (Zhang et al., 2024a) | 16015 | 4 | 180190 | 22523 | 22525 | 14822 | 768 | 14822 | 768 |

Table 6: Full results on the MKG-W and MKG-Y datasets.

| Method | MKG-W | | | | MKG-Y | | | |
|---|---|---|---|---|---|---|---|---|
| | **MRR** | **Hit1** | **Hit3** | **Hit10** | **MRR** | **Hit1** | **Hit3** | **Hit10** |
| **TransE** | 29.19 | 21.06 | 33.20 | 44.23 | 30.73 | 23.45 | 35.18 | 43.37 |
| **DistMult** | 20.99 | 15.93 | 22.28 | 30.86 | 25.04 | 19.33 | 27.80 | 35.95 |
| **ComplEx** | 24.93 | 19.09 | 26.69 | 36.73 | 28.71 | 22.26 | 32.12 | 40.93 |
| **RotatE** | 33.67 | 26.80 | 36.68 | 46.73 | 34.95 | 29.10 | 38.35 | 45.30 |
| **IKRL** | 32.36 | 26.11 | 34.75 | 44.07 | 33.22 | 30.37 | 34.28 | 38.26 |
| **TBKGC** | 31.48 | 25.31 | 33.98 | 43.24 | 33.99 | 30.47 | 35.27 | 40.07 |
| **TransAE** | 30.00 | 21.23 | 34.91 | 44.72 | 28.10 | 25.31 | 29.10 | 33.03 |
| **MMKRL** | 30.10 | 22.16 | 34.09 | 44.69 | 36.81 | 31.66 | 39.79 | 45.31 |
| **RSME** | 29.23 | 23.36 | 31.97 | 40.43 | 34.44 | 31.78 | 36.07 | 39.09 |
| **VBKGC** | 30.61 | 24.91 | 33.01 | 40.88 | 37.04 | 33.76 | 38.75 | 42.30 |
| **OTKGE** | 34.36 | 28.85 | 36.25 | 44.88 | 35.51 | 31.97 | 37.18 | 41.38 |
| **MoSE** | 33.34 | 27.78 | 33.94 | 41.06 | 36.28 | 33.64 | 37.47 | 40.81 |
| **IMF** | 34.50 | 28.77 | 36.62 | 45.44 | 35.79 | 32.95 | 37.14 | 40.63 |
| **QEB** | 32.38 | 25.47 | 35.06 | 45.32 | 34.37 | 29.49 | 36.95 | 42.32 |
| **VISTA** | 32.91 | 26.12 | 35.38 | 45.61 | 30.45 | 24.87 | 32.39 | 41.53 |
| **MANS** | 30.88 | 24.89 | 33.63 | 41.78 | 29.03 | 25.25 | 31.35 | 34.49 |
| **MMRNS** | 35.03 | 28.59 | 37.49 | 47.47 | 35.93 | 30.53 | 39.07 | 45.47 |
| **MoMoK** | 35.89 | 30.38 | 37.54 | 46.13 | 37.91 | 35.09 | 39.20 | 43.20 |

## C.2 EVALUATION PROTOCOLS

We conduct a link prediction task on the datasets, the mainstream MMKGC task. Following existing works, we use rank-based metrics like mean reciprocal rank (MRR) and Hit@K (K=1, 3, 10) to evaluate the results. Besides, we employ the filter setting in the prediction results to remove the candidate triples that already appeared in the training data for fair comparisons. The final results are the average of both head prediction and tail prediction. MRR and Hit@K can be calculated as:

$$\mathbf{MRR} = \frac{1}{|\mathcal{T}_{test}|} \sum_{i=1}^{|\mathcal{T}_{test}|} \left( \frac{1}{r_{h,i}} + \frac{1}{r_{t,i}} \right) \tag{17}$$

$$\mathbf{Hit@K} = \frac{1}{|\mathcal{T}_{test}|} \sum_{i=1}^{|\mathcal{T}_{test}|} \left( \mathbf{1}(r_{h,i} \leq K) + \mathbf{1}(r_{t,i} \leq K) \right) \tag{18}$$

where $r_{h,i}$ and $r_{t,i}$ are the results of head prediction and tail prediction respectively, $\mathcal{T}_{test}$ is the test triple set.

## C.3 ADDITIONAL EXPERIMENTAL RESULTS

We present the additional experimental results in this section.

### C.3.1 FULL RESULTS ON MKG-W AND MKG-Y

In our experiments, we present the MRR and Hit@1 results in Table 1. We now present the results for the complete set of four metrics in both datasets in the Table 6.

Table 7: MMKGC results on the FB15K-237 dataset.

| Method | MRR | Hit@1 | Hit@3 | Hit@10 |
|---|---|---|---|---|
| **IKRL*** | - | 19.4 | 28.4 | 45.8 |
| **TransAE*** | - | 19.9 | 31.7 | 46.3 |
| **RSME*** | - | 24.2 | 34.4 | 46.7 |
| **MoSE-AI*** | - | 25.5 | 37.6 | 51.8 |
| **MoSE-BI*** | - | 28.1 | 41.1 | 56.5 |
| **MoSE-MI*** | - | 26.8 | 39.4 | 54.0 |
| **QEB** | 29.93 | 20.73 | 33.37 | 48.03 |
| **AdaMF** | 32.56 | 23.33 | 35.85 | 51.13 |
| **MoMoK (Ours)** | 36.08 | 27.54 | 39.60 | 55.73 |

Table 8: Ablation study on the MKG-Y and KVC16K datasets. The results with * are from the MoSE paper (Zhao et al., 2022). - means the results are not reported in the given paper.

| Setting | MKG-Y | | | | KVC16K | | | |
|---|---|---|---|---|---|---|---|---|
| | MRR | Hit@1 | Hit@3 | Hit@10 | MRR | Hit@1 | Hit@3 | Hit@10 |
| **Full Model** | **37.91** | **35.09** | **39.20** | **43.20** | **16.87** | **10.53** | **18.26** | **29.20** |
| **(1.1) Structural Modality** | 35.66 | 32.14 | 37.61 | 41.51 | 15.66 | 9.52 | 16.95 | 27.54 |
| **(1.2) Image Modality** | 35.99 | 32.84 | 37.98 | 41.23 | 15.50 | 9.56 | 16.59 | 27.11 |
| **(1.3) Text Modality** | 35.73 | 32.41 | 37.76 | 41.18 | 15.56 | 9.61 | 16.68 | 27.16 |
| **(1.4) Joint Modality** | 35.83 | 32.14 | 37.81 | 42.60 | 16.45 | 10.22 | 17.61 | 28.59 |
| **(2.1) w/o relational temperature** | 37.17 | 33.93 | 38.66 | 42.85 | 16.74 | 10.31 | 18.16 | 29.01 |
| **(2.2) w/o noise $\delta_m$** | 37.63 | 34.62 | 39.11 | 43.11 | 16.77 | 10.40 | 18.08 | 29.07 |
| **(2.3) w/o adaptive fusion** | 36.54 | 33.05 | 38.17 | 42.28 | 16.55 | 10.28 | 17.69 | 28.70 |
| **(2.4) w/o joint training** | 35.33 | 31.69 | 37.01 | 40.99 | 15.92 | 9.69 | 17.20 | 27.91 |
| **(2.5) w/o ExID** | 36.37 | 32.56 | 38.64 | 42.70 | 16.05 | 9.87 | 17.24 | 28.05 |

These added experimental results allow us to draw further conclusions. For example, we can find that our method MoMoK outperforms baselines on MRR/Hit@1/Hit@3 on MKG-W and Hit@1/Hit@3 on MKG-Y. We can conclude that our method will perform better on fine-grained ranking metrics such as Hit@1 than on the coarse-grained ranking metric Hit@10. The trend reflected in the new results is the same as DB15K and KVC16K.

### C.3.2 Additional Results on FB15K-237

We present some more results on FB15K-237, shown in the following Table 7. These results indicate that MoMoK still performs well on classic MMKG benchmarks like FB15K-237. Only MoSE-BI is currently slightly better. Compared with recent baselines like AdaMF, MoMoK performs better. Based on our observation of the source code, MoSE achieves this result by setting the dimension of embedding to 2000, while our method MoMoK uses 250.

Compared with MoSE, MoMoK is an end-to-end training framework, MoSE needs to learn the parameters of the ensemble once more after the model is trained, which is also a feature of our design.

### C.3.3 Additional Ablation Studies

We present the experimental results of MKG-Y and KVC16K in the Table 8 and do a more in-depth analysis of the whole ablation study.

These experimental results are consistent with those shown in the paper for MKG-W and DB15K. That is, each module contributes to the final performance. Besides, we can also do some analysis from the new experimental results on this issue of the small impact of the different modules that you mentioned. On the one hand, we conducted a fine-grained ablation study, with specialized experiments on several key designs. On the other hand, we can observe that there is also a big difference in the contribution of the same module in different datasets. This is related to the dataset itself, in addition to the fact that the foundational elements of our entire framework, such as the TuckER score itself,

have better performance and subsequent designs have made significant enhancements based on this foundation.

## D   LIMITATIONS OF OUR WORK

Our main work is the design and implementation of a novel MMKGC framework. Of course, there are some limitations to our work. The main points are as follows:

- Limitations of task scenarios. Our research focuses on a specific research field called MMKGC and our method MOMOK is designed specifically for this task. We do not generalize this framework to more multi-modal tasks.
- Integration with LLM trends. Our approach uses a classical embedding-based approach in studying the MMKGC problem and does not combine the MMKG with the latest LLM trends in a synergistic way.
- Limitations of the experiment. Due to the lack of standard datasets for super large-scale experiments at MMKG, our experiments were conducted mainly on medium-sized datasets.

## E   BROADER IMPACTS OF OUR WORK

Our work focuses on reasoning about multi-modal knowledge, which can help us discover new possible associations in large-scale semantic networks and encyclopedic knowledge and expand existing encyclopedic knowledge bases such as Wikidata, etc. The positive social impact of our work is to help build and expand the Internet knowledge-sharing community, and to make more accumulation and deposition of linked data. We do not believe that our research will have a negative social impact, and we will also take active steps to avoid misuse of our methods.

