# OpenReview forum: "Multiple Heads are Better than One: Mixture of Modality Knowledge Experts for Entity Representation Learning"
_ICLR.cc/2025/Conference — ICLR 2025 Poster_

### Official Review · Reviewer_mwpc · 2024-10-20

**Soundness:** 3
**Presentation:** 4
**Contribution:** 2
**Rating:** 8
**Confidence:** 4

**Summary:**

The authors argue the limitations of entity-wise embedding fusion in existing MMKGC work and introduce a new method, MoMoK, which can learn relation-guided entity representations for each modality. The experimental results demonstrate the superiority and robustness of MoMoK.

**Strengths:**

The paper is well-written, and I was able to grasp authors' motivation and the general process of MoMoK at first glance. The algorithm is clearly presented overall. The performance of MoMoK surpassed existing methods by a significant margin. Sufficient experimental evidence supports the robustness of MoMoK.

**Weaknesses:**

I think **the discussion of the importance of KG Completion task, especially its position in the context of the pervasiveness of LLMs today, is lacking**. This reduces the overall interest and significance of the work from the audience's perspective, considering LLMs' strong performance in general knowledge reasoning. I suggest the authors add a brief discussion in the introduction or in the last section. Additionally, several places lack detail and need clarification (see questions).

**Questions:**

1. In the second paragraph of Section 4.1, are $\mathcal{W}\_{m,1}$, $\mathcal{W}\_{m,2}$, ... learnable matrics? What are their shapes? Also, what is the exact operation in $\mathcal{V}\_{m,i}^{e}=\mathcal{W}\_{m,i}(e\_m)$? If it is just a simple multiplication, I feel using parentheses is improper.
2. In Eq. (2), what is the purpose/function of the temperature $\varepsilon_r$?
3. In Eq. (3), I think the relation $r$ is missing in the components $P\_m(\hat{e}\_{m})$ and $P\_n(\hat{e}\_{n})$?
4. In line 266, where the joint score of each triple is treated as the sum of the scores across all modalities, did the authors try other designs, such as using only the '$J$' modality score? I am curious about the results of using only the '$J$' modality.
5. What's the setting of the number of experts (the value of $K$) in the main experiment? This should be described in the implementation details.
6. In Section 5.1, what is the difference between the MKG-W and MKG-Y datasets?

---

> ### Author Response · Authors · 2024-11-18
> **Official Rebuttal from Authors**
>
> Dear Reviewer:
>
> Thank you for your insightful feedback. Below are our responses to each of your comments and questions. Hope our response finds you well and improves your impression on our paper.
>
> ## About the design in Section 4.1
> Yes, $ W_{m, i} $are learnable projection layers (nn.Linear in PyTorch with bias). They are in the shape of (hidden state dimension, modality feature dimension), aiming to first project the raw modality features into a hidden space and obtain the multi-pespective embeddings $ Vm,i $of each entity. Since W is not only a linear layer but also contains bias, we use this representation.
>
> ## About Temperature in Equation (2)
> This temperature is a relation-wise parameter. For each relation, we set one learnable parameter as the temperature. It is used to change the experts's weights when facing different relation prediction tasks. As we introduced in Figure 1 and Section 3, multi-modal knowledge graph completion (MMKGC) is a triple-level task. For a given head entity h, when we predict under different relation r, the answer would be different. Therefore, we expect that the MMKGC model can learn **different entity representations under different relations** for better prediction. Therefore we design such a temperature that introduces the information of relations when learning entity representations. Compared to traditional methods that introduce relationship information only at the time of final prediction, our approach can be analogized as an early fusion of features.
>
>
>
> ## About Equation (3)
> Sorry for this typo and we have fixed it in the revised version of our paper.
>
> ## About the joint modality
> The issue of the use of Joint Score is specifically explored in the ablation study, as you can see in Table 3 of Section 5.5 we explored the performance of each modality individually for prediction, and from the results of this experiment, we can also see that adding all the modalities together will have the effect of From the results of this experiment, we can also see that adding all modalities together will result in a better model and that a model trained using any of the modalities alone (including joint modality) will not be as good as a full model.
>
>
>
> ## The setting of expert number
> This parameter is related to the design of Section 4.1. Our model has K experts in the ReMoKE module to obtain K different multi-perspective embeddings of each entity. We further make an exploration about the hyper-parameter K in Section 5.5. Thanks for pointing this out, we've added this section to the paper's implementation details.
>
>
>
> ## The difference between MKG-W and MKG-Y
> MKG-W and MKG-Y are two different datasets proposed in MMRNS. Both of them are constructed based on the OpenEA project. However, they are two different MMKGs because their data sources are different. The modality information is collected by search engines. MKG-W is based on Wikidata and MKG-Y is based on YAGO. Both of the data sources are famous open-source knowledge graphs.
>
>
>
> These are our responses to your comments and questions, and we hope that our responses will answer your concerns and improve your impression of our paper. We hope that you can improve the score of your decision, which will mean a lot to us.

---

> > ### Comment · Reviewer_mwpc · 2024-11-18
> > **Reply to The Authors**
> >
> > Thanks the authors for their responses.
> >
> > The authors have addressed all my questions and made the necessary corrections to the paper. I now consider all my concerns resolved and have increased my overall rating, as the paper appears to be solid work.
> >
> > I’ve also adjusted the breakdown ratings to reflect my evaluation more accurately.

---

> > > ### Author Response · Authors · 2024-11-19
> > > **Thank you!**
> > >
> > > Dear Reviewer:
> > >
> > > Thank you for your insightful comments and feedback!
> > >
> > > Sincerely,
> > >
> > > Authors

---

### Official Review · Reviewer_rUdt · 2024-11-04

**Soundness:** 3
**Presentation:** 3
**Contribution:** 2
**Rating:** 6
**Confidence:** 4

**Summary:**

This paper introduces a novel model named MOMOK (Mixture of Modality Knowledge Experts), designed to address the challenges in multimodal knowledge graph completion tasks. The innovation lies in the relation-guided multimodal expert networks, which dynamically adjust the weighting of modal information based on relation, enhancing the model's completion performance. The experimental results demonstrate that MOMOK surpasses existing baseline models across multiple public datasets, showcasing its effectiveness in handling multimodal information. The paper is well-writing with contributions in both theoretical design and empirical results.

**Strengths:**

1. The introduction of relation-guided multimodal expert networks with adaptive weighting mechanisms enables the model to fully exploit different perspectives within each modality based on relational context, which represents a novel and impactful focus.
2. By minimizing mutual information, the model effectively reduces redundant information between modalities, enhancing disentanglement among the experts. This is a particularly noteworthy design feature.
3. The ablation experiments validate the importance of each module in the model's performance, particularly highlighting the significance of joint training and information disentanglement, proving the rationale behind the design.
4. The paper presents five well-defined research questions (RQs) that thoroughly address the model’s strengths, performance in complex environments, and the contribution of each design module. This makes the experimental section well-structured and logically cohesive.

**Weaknesses:**

1. The novelty is somewhat limited. The paper claims existing works lack considering the relational context, and proposes MoMoK frameworks by incorporates relation-guided modality knowledge experts for each modality, which constructs expert networks in each modality. But in my opinion, there are also lots of prior work (such as MoSE) considering relational context.
2. It would be beneficial to introduce datasets used in baselines for comprehensive comparison. The baselines, such as MoSE, further conduct experiments on (FB-15K, WN18, WN9) which this paper did not used. Including these datasets would be helpful to further demonstrate the robustness of the proposed model.
3. The expert networks rely on raw embeddings, which may limit their ability to fully capture the diversity within each modality. This could lead to less distinct perspectives being generated by the experts, especially if the raw embeddings have already compressed or lost key details, thereby reducing the overall effectiveness of the expert network design.
4. While the paper introduces CLUB to estimate the upper bound of mutual information and designs a corresponding loss function, it lacks sufficient explanation for why the conditional probabilities of different entities are used for disentanglement in formula 8.


Other Minor Weaknesses:

1. The related work section is too brief and fails to emphasize the superiority of the proposed framework compared to existing approaches. For example, in the experimental section, you mentioned the similarities between your method and MoSE and IMF as baselines, but you did not provide a more detailed discussion of the differences. This could be elaborated in the related work section to clarify the distinctions between your approach and these existing methods.
2. The paper discusses the use of three modalities, image, text, and structure
but it appears to lack a clear explanation regarding how the raw embeddings for the structural modality are obtained. Neither the main body of the paper nor the appendix provides detailed information about the specific model or method used to generate raw embeddings for the structural modality.

Typos:

1. Page2 'constrative' maybe is 'contrastive'
2. The second-best method in the MRR indicator of the MKG-W dataset in Table 2 should be MMRNS (35.03), but it was incorrectly marked as IMF (34.50).

**Questions:**

1. Given that the expert networks rely on raw embeddings, how does the model ensure that sufficient diversity within each modality is captured? Are there mechanisms in place to mitigate the potential loss of key details due to the use of compressed embeddings?
2. Could you provide further clarification on the rationale behind using the conditional probabilities of different entities for disentanglement in formula 8? How does this approach ensure effective disentanglement between experts?
3. In the final case study, the authors visualize the weighting between modalities. How does the model distinguish and prioritize features within a single modality?

---

> ### Author Response · Authors · 2024-11-18
> **Official Rebuttal from Authors**
>
> Dear Reviewer:
>
> Thank you for your insightful feedback. We have made a response to each of your comments and questions. Hope our response finds you well and improves your impression on our paper.
>
> ## Novelty of our method
> In KGC prediction, relation context is something that needs to be taken into account, and the vast majority of traditional approaches are to train relation embeddings individually and apply these relation embeddings to the calculation of scores. While MoSE is an ensemble learning MMKGC method, its underlying backbone is the classic ComplEx model, and the relation context is considered when doing the ensemble. This consideration of the relation context remains at the final stage. The entity embeddings of these methods would not be changed when facing different relation contexts. The relation context only appears in the final step of these models.
>
>
>
> However, our proposed framework, MoMoK considers the relational context in a more early stage when obtaining the entity representation. We achieve this goal by the design of relation-guided modality knowledge experts. In the subsequent score calculation, the relation context will still participate in the score calculation in the form of embeddings. In this way, we consider the relation context in both phases of the model, which also makes our entity embedding learning can change dynamically with different relation contexts during both the training and inference stages, which is one of our novelties.
>
>
>
> ## More results on more datasets
>
>
> We find this question valuable and therefore also complement the results of our approach on a more standard dataset. Current research on uni-modal KGC usually employs FB15K-237 and WN18RR for evaluation. Other datasets like FB15K / FB-IMG / WN18 have certain problems that result in lesser use. It is because there are some trivial patterns and inverse relations [1, 2] in these datasets that are easy to be learned by the models, making these datasets lack of generalization and challenge, leading to very high results for most of the models and making it difficult to differentiate between the capabilities of different models. According to our attempts, the WordNet series of datasets have higher-order abstractions in their corresponding image information because their entities are words, making these images less useful for MMKGC prediction. At the same time, this part of the information is also more difficult to obtain, and the download methods provided in some open-source repositories require more complicated configurations to obtain the full images.
>
> Therefore, we present some more results on FB15K-237, shown in the following Table. The results with * are from the MoSE paper. - means the results are not reported in the given paper.
>
> | **Method** | MRR | Hit@1 | Hit@3 | Hit@10 |
> | :---: | :---: | :---: | :---: | :---: |
> | **IKRL*** | - | 19.4 | 28.4 | 45.8 |
> | **TransAE*** | - | 19.9 | 31.7 | 46.3 |
> | **RSME*** | - | 24.2 | 34.4 | 46.7 |
> | **MoSE-AI*** | - | 25.5 | 37.6 | 51.8 |
> | **MoSE-BI*** | - | 28.1 | 41.1 | 56.5 |
> | **MoSE-MI*** | - | 26.8 | 39.4 | 54.0 |
> | **QEB** | 29.93 | 20.73 | 33.37 | 48.03 |
> | **AdaMF** | 32.56 | 23.33 | 35.85 | 51.13 |
> | **MoMoK (Ours)** | 36.08 | 27.54 | 39.60 | 55.73 |
>
>
> These results indicate that MoMoK still performs well on classic MMKG benchmarks like FB15K-237. Only MoSE-BI is currently slightly better. Compared with recent baselines like AdaMF, MoMoK performs better. **Based on our observation of the source code, MoSE achieves this result by setting the dimension of embedding to 2000, while our method MoMoK uses 250.**
>
> Compared with MoSE, MoMoK is an end-to-end training framework, MoSE needs to learn the parameters of the ensemble once more after the model is trained, which is also a feature of our design.
>
>
>
> [1] Convolutional 2D Knowledge Graph Embeddings
>
>
>
> [2] Observed Versus Latent Features for Knowledge Base and Text Inference

---

> ### Author Response · Authors · 2024-11-18
> **Official Rebuttal from Authors (2)**
>
> ## The design of expert network (also w/ Q1)
> The expert networks treat the raw modality embeddings as the inputs and produce relation-aware entity embeddings as outputs. In this process, we use project layers (W) and mixture-of-experts architecture to capture the representative information in the given embeddings. Note that it is a common setting for embedding-based MMKGC methods, which means that all the MMKGC baselines we used in the experiments follow such a setting and use the raw modality information as inputs with further process pipelines. From the experiment results, we can observe that MoMoK does well in leveraging the existing information in the raw embeddings compared with baseline methods.
>
> But your concern is still valuable, how to fully unleash the power of raw modality data (images and texts) is another significant topic in multi-modal representation learning, which will be our future direction.
>
> To solve the problem you mentioned in Question 1, our MoMoK framework employs multiple modality experts in each modality, which can learn some important and key features from diverse perspectives. We design tunable Gaussian noise and relation guidance to enhance this module. The tunable Gaussian noise is a classic design in MoE [3] which can guide sparsification learning for experts in MoE. The relation guidance is our new design considering the realities of the MMKG scenario.
>
>
>
> [3] Outrageously large neural networks: The sparsely-gated mixture-of-experts layer.
>
> ## More about the CLUB loss in Equation 8 (also w/ Q2)
> Equation 8 is our core design for expert information disentanglement. Note that we have set multiple experts in each modality. We expect these experts can learn intra-modality knowledge from different perspectives. Therefore, we design such a module to guide different experts to learn different aspects by considering the mutual information. Mutual information is an important concept in information theory and modern deep learning, which can measure the dependence of two variables.
>
> Famous representation learning technologies, such as InfoNCE, are widely used to maximize mutual information. Maximizing mutual information enhances the dependence and correlation of two variables while minimizing mutual information allows two variables to be decoupled from each other. In our design, we choose to disentangle the experts. Therefore, we employ CLUB, a famous mutual information minimizing framework in our design. We provide a more detailed theoretical analysis in the Appendix to present our opinion.
>
>
>
> ## More related work analysis
> Due to the page limitations of the submission, we did not provide an in-depth analysis of the differences between the different MMKGC methods in the related works of the manuscript. However, we have made a preliminary delineation in the introductory section of the baselines of the Appendix. We'll refine this part in the revised submission. For this question of your concern about the difference between MoMoK and MoSE, we've organized it with this table below:
>
>
>
> | Method | Ensemble | Multi-modal Fusion | Relation-aware Fusion/Ensemble? | Intra-modality   Disentangle |
> | :---: | :---: | :---: | :---: | :---: |
> | MoSE | Yes | No | Yes (Final Decision) | No |
> | IMF | Yes | Yes | No | No |
> | MoMoK | Yes | Yes | Yes (Entity Embedding) | Yes |
>
>
>
>
> Though MoSE, IMF, and our method MoMoK are all ensemble-based MMKGC methods, MoMoK differs in many ways from the two existing methods. Compared with MoSE and IMF, MoMoK has a relation-aware multi-modal fusion module (ReMoKEs) and an intra-modality disentanglement module (ExID), which makes MoMoK outperform the baselines according to our experiments.
>
>
>
> **Note that we have revised our paper and added more related works analysis in the Appendix A.**
>
>
>
> ## The learning of structural embedding
> In our design, the structural embeddings are learned during training. This is different from the way raw embedding is handled for image and text, where the raw embedding for these modalities is pre-extracted, whereas structural embedding is learned in training after random initialization. This is a common setting in MMKGC models.
>
>
>
> ## Typos
>
>
> Sorry for the two typos, we have fixed it in our revised version.

---

> ### Author Response · Authors · 2024-11-18
> **Official Rebuttal from Authors (3)**
>
> ## About visualization (w/ Q3)
> The phenomenon you mention is related to the design of our model. The reason has two main parts, within each modality expert, we form a differentiation within each modality through MoE design, so that each expert learns different contents, and between modalities, we again highlight the weight of the key modalities through the dynamic modal feature fusion, so that the synergistic effect of the two bootstrapping makes the model internal for the entity's modal weights appear distributed as shown in the visualization results.
>
> And we plot the adaptive weight distribution within a single modality and across multiple modalities in the visualization results. We can see that not only do the experts within a modality form a differentiation, but the weights of the modalities to each other are also significantly different in different RELATION contexts.
>
>
>
> These are our responses to your comments and questions, and we hope that our responses will answer your concerns and improve your impression of our paper. We hope that you can improve the score of your review, which will mean a lot to us.

---

> > ### Comment · Reviewer_rUdt · 2024-11-27
> >
> > Thanks to the authors for their responses. I would like to maintain my score as it has been positive.

---

> > > ### Author Response · Authors · 2024-11-27
> > > **Thank you!**
> > >
> > > Dear Reviewer:
> > >
> > > Thank you for your feedback!

---

### Official Review · Reviewer_Diph · 2024-11-04

**Soundness:** 3
**Presentation:** 4
**Contribution:** 3
**Rating:** 8
**Confidence:** 3

**Summary:**

This paper proposes a multi-modal mixture-of-experts entity representation framework MoMoK for knowledge graph completion task. MoMoK consists of three different modules called relation-guided modality knowledge experts, multi-modal joint decision, and expert information disentanglement.

**Strengths:**

- Multi-modal knowledge graph representation learning is a hot topic in the KG community.
- The three modules in the design of the method proposed by the authors are self-consistent and the authors provide a more detailed theoretical proof.
- The experimental part of the workload is sufficient, and the authors have done more experiments to prove the effectiveness, robustness, and efficiency of the method. Detailed case study and visualization results are also provided.

**Weaknesses:**

- The discussion of MAIN RESULTS needs to go further, have the authors explored in depth why MKG-Y has a lesser MRR than baseline AdaMF?
- The fonts of the figure3 and figure4 are not consistent.
- The detailed settings of Section5.4 should be more carefully introduced. About how the training datasets are built and the detailed evaluation protocol.

**Questions:**

- Please make a response to the comments in weaknesses.
- What exactly are the advantages of the traditional embedding-based approach over the LLM-based approach for knowledge graph completion?

---

> ### Author Response · Authors · 2024-11-18
> **Official Rebuttal from Authors**
>
> Dear Reviewer:
>
> Thank you for your insightful feedback. We have made a response to each of your comments and questions. Hope our response finds you well and improves your impression on our paper.
>
> ## More exepriment analysis
>
>
> In our experiments, we present the MRR and Hit@1 results in Table 2. We now present the results for the complete set of four metrics in both datasets in the table below:
>
> | **Method** | **MKG-W** | | | |
> | :---: | :---: | --- | --- | --- |
> | | **MRR** | **Hit1** | **Hit3** | **Hit10** |
> | **TransE** | 29.19 | 21.06 | 33.20 | 44.23 |
> | **DistMult** | 20.99 | 15.93 | 22.28 | 30.86 |
> | **ComplEx** | 24.93 | 19.09 | 26.69 | 36.73 |
> | **RotatE** | 33.67 | 26.80 | 36.68 | 46.73 |
> | **IKRL** | 32.36 | 26.11 | 34.75 | 44.07 |
> | **TBKGC** | 31.48 | 25.31 | 33.98 | 43.24 |
> | **TransAE** | 30.00 | 21.23 | 34.91 | 44.72 |
> | **MMKRL** | 30.10 | 22.16 | 34.09 | 44.69 |
> | **RSME** | 29.23 | 23.36 | 31.97 | 40.43 |
> | **VBKGC** | 30.61 | 24.91 | 33.01 | 40.88 |
> | **OTKGE** | 34.36 | 28.85 | 36.25 | 44.88 |
> | **MoSE** | 33.34 | 27.78 | 33.94 | 41.06 |
> | **IMF** | 34.50 | 28.77 | 36.62 | 45.44 |
> | **QEB** | 32.38 | 25.47 | 35.06 | 45.32 |
> | **VISTA** | 32.91 | 26.12 | 35.38 | 45.61 |
> | **MANS** | 30.88 | 24.89 | 33.63 | 41.78 |
> | **MMRNS** | 35.03 | 28.59 | 37.49 | 47.47 |
> | **MoMoK  (Ours)** | 35.89 | 30.38 | 37.54 | 46.13 |
>
>
>
>
> | **Method** | **MKG-Y** | | | |
> | :---: | :---: | --- | --- | --- |
> | | **MRR** | **Hit1** | **Hit3** | **Hit10** |
> | **TransE** | 30.73 | 23.45 | 35.18 | 43.37 |
> | **DistMult** | 25.04 | 19.33 | 27.80 | 35.95 |
> | **ComplEx** | 28.71 | 22.26 | 32.12 | 40.93 |
> | **RotatE** | 34.95 | 29.10 | 38.35 | 45.30 |
> | **IKRL** | 33.22 | 30.37 | 34.28 | 38.26 |
> | **TBKGC** | 33.99 | 30.47 | 35.27 | 40.07 |
> | **TransAE** | 28.10 | 25.31 | 29.10 | 33.03 |
> | **MMKRL** | 36.81 | 31.66 | 39.79 | 45.31 |
> | **RSME** | 34.44 | 31.78 | 36.07 | 39.09 |
> | **VBKGC** | 37.04 | 33.76 | 38.75 | 42.30 |
> | **OTKGE** | 35.51 | 31.97 | 37.18 | 41.38 |
> | **MoSE** | 36.28 | 33.64 | 37.47 | 40.81 |
> | **IMF** | 35.79 | 32.95 | 37.14 | 40.63 |
> | **QEB** | 34.37 | 29.49 | 36.95 | 42.32 |
> | **VISTA** | 30.45 | 24.87 | 32.39 | 41.53 |
> | **MANS** | 29.03 | 25.25 | 31.35 | 34.49 |
> | **MMRNS** | 35.93 | 30.53 | 39.07 | 45.47 |
> | **MoMoK  (Ours)** | 37.91 | 35.09 | 39.20 | 43.20 |
>
>
>
>
> We can draw more conclusions from these added experimental results. We can find that our method MoMoK outperforms baselines on MRR/Hit@1/Hit@3 on MKG-W and Hit@1/Hit@3 on MKG-Y. We can find that our method will perform better on fine-grained ranking metrics such as Hit@1 than on the coarse-grained ranking metric Hit@10. The trend reflected from the new results is the same as DB15K and KVC16K. Since MRR is calculated as an average of the inverse of all the ranks, it will be affected by both the coarse and fine ranking capabilities of the model.

---

> ### Author Response · Authors · 2024-11-18
> **Official Rebuttal from Authors (2)**
>
> ## Figure font
> Sorry for this typo, we have fixed it in the revised version of our paper. You can find it in the uploaded PDF of our paper.
>
>
>
> ## Detailed setting of Section 5.4
> In Section 5.4, we make a further exploration of the MMKGC performance under more complex scenarios. We set three different kinds of scenarios: modality noisy scenario, modality missing scenario, and link sparse scenario. Each of these three scenarios has a different setting.
>
> - Modality noisy scenario. Modality information for a certain ratio of entities is added with the noise sampled from a normal distribution.
> - Modality missing scenario. Modality information for a certain ratio of entities is missing.
> - Link sparse scenario. The triple in the training set is reduced by a certain percentage, making the data sparser.
> -
> For all of the three scenarios, we set a ratio (noisy rate, missing rate, and sparse rate) when conducting the experiments. The noise rate is set in 10%, 20%, 30%, ......, 90%. The missing rate is set from 10% to 50% with a margin of 10%. The sparse rate is set from 5% to 25% with a margin of 5%. For a given ratio, we select the same group of entities/triples and add modifications to them. For evaluation, we follow the same protocols as the main experiments and draw some line graphs to represent the MRR and Hit@10 results under the mentioned settings.
>
>
>
> ## Advantage of embedding-based MMKGC
> Although there are gradually more LLM-based KGC methods nowadays, these kinds of methods and the embedding-based methods we explore in this paper are two developmental routes. LLM-based methods rely entirely on textual information about entities and relations and utilize the powerful textual comprehension and generation capabilities of LLM for text-based prediction. These kinds of methods are very useful for knowledge. This type of approach is less capable of utilizing the rich graph structure information present in the graph, and also requires fine-tuning the LLM, resulting in a **higher cost for its training and inference**.
>
>
>
> One of the advantages of the embedding-based approach is the high efficiency of training and inference, and the second is the more flexible utilization of modal information, which only requires the input of modal eigenvectors of the entity for prediction, and does not strictly require the presence of explicit text. In the case of modal information is partially missing or noisy, it still has better performance. Of course, the combination of embedding-based and LLM-based methods will be the future development direction, and the combination of Knowledge Graph and LLM will also be an important research direction.
>
>
>
> Hope that our response finds you well and can improve your assessment of our paper. We would greatly appreciate it if you could consider raising your score.
>
>
>
> These are our responses to your comments and questions, and we hope that our responses will answer your concerns and improve your impression of our paper. We hope that you can improve the score of your decision, which will mean a lot to us.

---

> > ### Comment · Reviewer_Diph · 2024-11-27
> >
> > Thanks to the authors for their responses, the responses address my concerns.

---

> > > ### Author Response · Authors · 2024-11-27
> > > **Thank you !**
> > >
> > > Dear Reviewer:
> > >
> > > We'd like to thank you for your insightful feedback. We will continue to improve our paper.
> > >
> > > Sincerely,
> > >
> > > Authors

---

### Official Review · Reviewer_mPTp · 2024-11-06

**Soundness:** 3
**Presentation:** 3
**Contribution:** 3
**Rating:** 6
**Confidence:** 4

**Summary:**

This paper proposes a novel framework, Mixture of Modality Knowledge Experts (MoMoK), which aims to improve the quality of multimodal entity representation learning through relation guidance and expert information decoupling in multimodal knowledge graphs (MMKGs). MoMoK achieves better triple prediction by designing a relation-guided modal knowledge expert network, adaptively aggregating multi-view embeddings, and adopting a multimodal joint decision mechanism. Experimental results show that MoMoK performs well on four public MMKG benchmarks and significantly outperforms existing methods.

**Strengths:**

1. This paper considers the modal information hidden in different relational contexts and designs a relation-guided modal knowledge expert network to adaptively aggregate multi-view embeddings.
2. On four public MMKG benchmarks, MoMoK significantly outperforms recent baseline methods and demonstrates its superior performance in complex scenarios.
3. Through detailed ablation analysis, this paper verifies the effectiveness of the design and provides intuitive case studies to enhance the interpretability of the model.

**Weaknesses:**

1. Table 2 lacks a comparison of the number of model parameters. Although the authors have discussed the training memory consumption and time efficiency of some methods in Section 5.6, it is beneficial to supplement these values for all baseline models in Table 2 for fair comparison.
2. The description in Table 4 does not seem to be that comprehensive. Does the GPU memory here refer to the memory during inference or the memory during training? Does the time in seconds represent the time it takes to infer one sample?

**Questions:**

1. The parameter \lambda seems small enough. Can you explain why club loss can improve the model by about one point in almost every metric under a weight of 1e-4 (Table 3)?
2. The method seems to be insufficiently integrated with the multimodal large model. This paper uses BERT and VGG to extract the original features of the model. Have you tried to use a multimodal large model to represent them uniformly? I would like to know the performance comparison between the embedding representation of the multimodal large model and MoMoK.

---

> ### Author Response · Authors · 2024-11-18
> **Official Rebuttal from Authors**
>
> Dear Reviewer:
>
> Thank you for your insightful feedback. We have made a response to each of your comments and questions. Hope our response finds you well and improves your impression on our paper.
>
> ## Model parameter size analysis
> We think this comment is very valuable, so we have dutifully analyzed further model parameter quantities.
>
> - For traditional uni-modal KGC methods, their core parameters focus on the embedding of the entities and relations. Each embedding is a d-dimensional vector.
> - For multi-modal KGC methods, they have additional parameters including raw modality features ands pecific multi-modal fusion modules.
> - For negative sampling methods, they have additional parameters in the negative sampling module.
> -
> At the same time, the number of parameters can vary greatly from one method to another due to differences in their design. Next, we counted the number of parameters for different models on the DB15K dataset with d = 250. We used the same few methods from Table 4 to make a fairer comparison.
>
>
>
> | Method | #Parameters | Time | GPU Memory |
> | :---: | :---: | :---: | :---: |
> | MMKRL | 67.1M | 7.5s | 4504MB |
> | OTKGE | 71.8M | 70.1s | 2540MB |
> | MMRNS | 26.4M | 25.5s | 25582MB |
> | MoMoK (Ours) | 79.0M | 9.8s | 5900MB |
>
>
>
>
> From this table, we can find that our method MoMoK consists of a few more parameters compared with MMKRL and OTKGE. Though recent SOTA methods like OTKGE and MMRNS have fewer parameters, they design very complex computations and processes on their modules. For example, OTKGE implements an optimal transport algorithm in its model and MMRNS designs complex negative sampling strategies. These designs can greatly affect the training efficiency and GPU memory usage. Considering the performance, efficiency, GPU memory, and parameter number, the performance progress and efficiency changes achieved by our method are acceptable.
>
>
>
> ## More about Table 4
> In Table 4, the GPU memory and time cost refers to the training stage. We compared our method with several recent baselines to demonstrate that our method is time and space-efficient during the training time. This comparison is also made because MMKGC models tend to have more complex designs in the training phase, such as adversarial training (MMKRL), optimal transport (OTKGE), adaptive negative sampling (MMRNS), etc., which can significantly slow down the training of the model. We demonstrate the efficiency of our approach through this experiment. The time cost refers to the training time for 1 epoch.
>
>
>
> ## The loss weight \lambda
> In our design, we add a weight $ \lambda $to the ExID module for the CLUB loss. This loss aims to disentangle the mutual information in the modality expert networks, serving as an auxiliary loss. The current choice of parameters is an empirical conclusion that if the weights are set too large, they will affect the main training goal of the MMKGC, whereas a smaller number can give an effective guide to the model training. The nature of this loss is similar to the classical contrastive loss, which is used for the estimation of mutual information, although they have different end goals; contrastive loss aims at maximizing mutual information, while CLUB loss aims at minimizing mutual information. From this point of view we can also give an explanation that mutual information between different exerts should guarantee a certain degree of disentanglement, but outright disentanglement instead has no gain in performance.

---

> ### Author Response · Authors · 2024-11-18
> **Official Rebuttal from Authors (2)**
>
> ## Intergration of Larger Pre-trained Multi-modal Models
> In our paper,  the dataset we used is more diverse, and they used several different backbone models in obtaining text and image embedding of entities. And BERT and VGG are representative examples of them. We detail the dimensions of image and text embedding used for each data in Table, while the feature dimensions are different for different datasets and the pre-trained model used is also different, which we summarize below:
>
> | Dataset | Image Backbone | Text Backbone |
> | :---: | :---: | :---: |
> | MKG-W | BEiT | Sentence-BERT |
> | MKG-Y | BEiT | Sentence-BERT |
> | DB15K | VGG | BERT |
> | KVC16K | ViT | BERT |
>
>
> We can find a rich variety of backbones used in the course of our experiments. Our approach is an in-depth extension of the traditional embedding-based approach, while the other class of KGC methods is the one you mentioned, based on large multi-modal models for fine-tuning and similarity comparison, which requires training of the large multi-modal model itself. Often, they require very much computational resources and the results of training and inference are slow.
>
> Nowadays, the more mature multi-modal model-based MMKGC methods are MKGformer [1] and SGMPT [2]. We report the results on the FB15K-237 dataset. We present the results in the following Table. The results with * are from the original paper.
>
> | **Method** | MRR | Hit@1 | Hit@3 | Hit@10 |
> | :---: | :---: | :---: | :---: | :---: |
> | **VisualBERT*** | - | 21.7 | 32.4 | 43.9 |
> | **ViLBERT*** | - | 23.3 | 33.5 | 45.7 |
> | **MKGformer*** | - | 25.6 | 36.7 | 50.4 |
> | **SGMPT*** | - | 25.2 | 37.0 | 51.0 |
> | **MoMoK (Ours)** | **36.08** | **27.54** | **39.60** | **55.73** |
>
>
> From the experimental results, we can also see that our method performs better in all aspects of the metrics compared to these methods that directly use LARGE multi-modal models, and at the same time, from the information provided in the original paper, the parameter sizes of these models are around 950M and take close to 10 hours to train. This is also low compared to the efficiency we show in Table 4, so our **MoMoK does better in terms of performance and efficiency** compared to such methods. Of course, this will also be our future direction.
>
> These are our responses to your questions. We hope that our response will solve your concerns and improve your rating of our paper, and we would greatly appreciate it if you could consider improving your score.
>
>
>
> [1] Hybrid Transformer with Multi-level Fusion for Multimodal Knowledge Graph Completion
>
>
>
> [2] Structure Guided Multi-modal Pre-trained Transformer for Knowledge Graph Reasoning
>
>
>
> These are our responses to your comments and questions, and we hope that our responses will answer your concerns and improve your impression of our paper. We hope that you can improve the score of your decision, which will mean a lot to us.

---

### Official Review · Reviewer_VfNT · 2024-11-07

**Soundness:** 2
**Presentation:** 2
**Contribution:** 2
**Rating:** 5
**Confidence:** 4

**Summary:**

This paper tackles MKGE problems by proposing a Mixture of Modality Knowledge Experts framework. The authors develop relation-guided modality knowledge experts to obtain relation-aware modality embeddings while reducing mutual information among the experts. Empirical studies validate the model's effectiveness.

**Strengths:**

1. Experiments in complex environments demonstrate MoMoK's resilience to noise, data sparsity, and missing modalities.
2. This work introduces a novel expert information disentanglement (ExID) module to separate expert decisions for each modality.
3. The rationale for designing the relation-aware MKGC method is evident.

**Weaknesses:**

1. Metrics Reporting: Table 2 displays only the Hit@1 and MRR metrics for MKG-W and MKG-Y, showing modest performance improvements for MoMoK. Please include additional metrics for MoMoK, AdaMF, and MMRNS to better assess the reported gains.

2. MoMoK on Classic MMKGC Datasets: Why wasn't MoMoK evaluated on classic MMKGC datasets like WN9-IMG, FB-IMG, WN18-IMG, and FB15K-237-IMG?

3. Modal Data and Encoder Details: The notation $\mathcal{X}_m(e)$ indicates information from multiple modalities, yet four datasets are used. Please clarify the specific modalities for each dataset, how multiple images are handled, and whether VGG and BERT are consistently used as encoders.

4. Notation Standardization: Some formulas, such as $\mathcal{E}_h$ in formula (1), could benefit from standardized notation to improve clarity and consistency.

5. Relation-Aware Temperature ($\mathcal{E}_r$): The relation-aware temperature $\mathcal{E}_r$ is important for GFN computation, but its definition is absent. Please provide its formal definition and its impact on model performance.

6. Score Function Comparison: The proposed method employs Tucker decomposition as its score function, while the baseline methods utilize various other functions.

7. Ablation Study: Ablation experiments indicate that components like adaptive fusion, noise $\delta_{r}$, and relational $\epsilon_{m}$ may have minimal individual impacts. Further analysis or emphasis on their significance in different experimental contexts is needed.

8. Ablation Experiments for MKG-Y and KVC16K: Can MKG-Y and KVC16K datasets be included in the ablation experiments? This would enhance the assessment of the method's performance across diverse data and conditions.

**Questions:**

please see above.

---

> ### Author Response · Authors · 2024-11-18
> **Official Rebuttal from Authors (1)**
>
> Dear Reviewer:
>
> Thank you for your insightful feedback. We have made a response to each of your comments and questions. Hope our response finds you well and improves your impression on our paper.
>
> ## More results on MKG-W and MKG-Y
> In our experiments, we present the MRR and Hit@1 results in Table 2. We now present the results for the complete set of four metrics in both datasets in the table below:
>
> | **Method** | **MKG-W** | | | |
> | :---: | :---: | --- | --- | --- |
> | | **MRR** | **Hit1** | **Hit3** | **Hit10** |
> | **TransE** | 29.19 | 21.06 | 33.20 | 44.23 |
> | **DistMult** | 20.99 | 15.93 | 22.28 | 30.86 |
> | **ComplEx** | 24.93 | 19.09 | 26.69 | 36.73 |
> | **RotatE** | 33.67 | 26.80 | 36.68 | 46.73 |
> | **IKRL** | 32.36 | 26.11 | 34.75 | 44.07 |
> | **TBKGC** | 31.48 | 25.31 | 33.98 | 43.24 |
> | **TransAE** | 30.00 | 21.23 | 34.91 | 44.72 |
> | **MMKRL** | 30.10 | 22.16 | 34.09 | 44.69 |
> | **RSME** | 29.23 | 23.36 | 31.97 | 40.43 |
> | **VBKGC** | 30.61 | 24.91 | 33.01 | 40.88 |
> | **OTKGE** | 34.36 | 28.85 | 36.25 | 44.88 |
> | **MoSE** | 33.34 | 27.78 | 33.94 | 41.06 |
> | **IMF** | 34.50 | 28.77 | 36.62 | 45.44 |
> | **QEB** | 32.38 | 25.47 | 35.06 | 45.32 |
> | **VISTA** | 32.91 | 26.12 | 35.38 | 45.61 |
> | **MANS** | 30.88 | 24.89 | 33.63 | 41.78 |
> | **MMRNS** | 35.03 | 28.59 | 37.49 | 47.47 |
> | **MoMoK (Ours)** | 35.89 | 30.38 | 37.54 | 46.13 |
>
>
>
> | **Method** | **MKG-Y** | | | |
> | :---: | :---: | --- | --- | --- |
> | | **MRR** | **Hit1** | **Hit3** | **Hit10** |
> | **TransE** | 30.73 | 23.45 | 35.18 | 43.37 |
> | **DistMult** | 25.04 | 19.33 | 27.80 | 35.95 |
> | **ComplEx** | 28.71 | 22.26 | 32.12 | 40.93 |
> | **RotatE** | 34.95 | 29.10 | 38.35 | 45.30 |
> | **IKRL** | 33.22 | 30.37 | 34.28 | 38.26 |
> | **TBKGC** | 33.99 | 30.47 | 35.27 | 40.07 |
> | **TransAE** | 28.10 | 25.31 | 29.10 | 33.03 |
> | **MMKRL** | 36.81 | 31.66 | 39.79 | 45.31 |
> | **RSME** | 34.44 | 31.78 | 36.07 | 39.09 |
> | **VBKGC** | 37.04 | 33.76 | 38.75 | 42.30 |
> | **OTKGE** | 35.51 | 31.97 | 37.18 | 41.38 |
> | **MoSE** | 36.28 | 33.64 | 37.47 | 40.81 |
> | **IMF** | 35.79 | 32.95 | 37.14 | 40.63 |
> | **QEB** | 34.37 | 29.49 | 36.95 | 42.32 |
> | **VISTA** | 30.45 | 24.87 | 32.39 | 41.53 |
> | **MANS** | 29.03 | 25.25 | 31.35 | 34.49 |
> | **MMRNS** | 35.93 | 30.53 | 39.07 | 45.47 |
> | **MoMoK   (Ours)** | 37.91 | 35.09 | 39.20 | 43.20 |
>
>
> These added experimental results allow us to draw further conclusions. For example, we can find that our method MoMoK outperforms baselines on MRR/Hit@1/Hit@3 on MKG-W and Hit@1/Hit@3 on MKG-Y.
>
> We can conclude that our method will **perform better on fine-grained ranking metrics** such as Hit@1 than on the coarse-grained ranking metric Hit@10. The trend reflected in the new results is the same as DB15K and KVC16K.
>
>
>
> ## More results on other datasets
> We find this question valuable and therefore also complement the results of our approach on a more standard dataset. Current research on uni-modal KGC usually employs FB15K-237 and WN18RR for evaluation. Other datasets like FB15K / FB-IMG / WN18 have certain problems that result in lesser use. It is because there are some trivial patterns and inverse relations [1, 2] in these datasets that are easy to be learned by the models, making these datasets lack of generalization and challenge, leading to very high results for most of the models and making it difficult to differentiate between the capabilities of different models. According to our attempts, the WordNet series of datasets have higher-order abstractions in their corresponding image information because their entities are words, making these images less useful for MMKGC prediction. At the same time, this part of the information is also more difficult to obtain, and the download methods provided in some open-source repositories require more complicated configurations to obtain the full images.
>
> Therefore, we present some more results on FB15K-237, shown in the following Table. The results with * are from the MoSE paper. - means the results are not reported in the given paper.
>
> | **Method** | MRR | Hit@1 | Hit@3 | Hit@10 |
> | :---: | :---: | :---: | :---: | :---: |
> | **IKRL*** | - | 19.4 | 28.4 | 45.8 |
> | **TransAE*** | - | 19.9 | 31.7 | 46.3 |
> | **RSME*** | - | 24.2 | 34.4 | 46.7 |
> | **MoSE-AI*** | - | 25.5 | 37.6 | 51.8 |
> | **MoSE-BI*** | - | 28.1 | 41.1 | 56.5 |
> | **MoSE-MI*** | - | 26.8 | 39.4 | 54.0 |
> | **QEB** | 29.93 | 20.73 | 33.37 | 48.03 |
> | **AdaMF** | 32.56 | 23.33 | 35.85 | 51.13 |
> | **MoMoK (Ours)** | 36.08 | 27.54 | 39.60 | 55.73 |
>
>
> These results indicate that MoMoK still performs well on classic MMKG benchmarks like FB15K-237. Only MoSE-BI is currently slightly better. Compared with recent baselines like AdaMF, MoMoK performs better. **Based on our observation of the source code, MoSE achieves this result by setting the dimension of embedding to 2000, while our method MoMoK uses 250.**

---

> ### Author Response · Authors · 2024-11-18
> **Official Rebuttal from Authors (2)**
>
> Compared with MoSE, MoMoK is an end-to-end training framework, MoSE needs to learn the parameters of the ensemble once more after the model is trained, which is also a feature of our design.
>
>
>
> [1] Convolutional 2D Knowledge Graph Embeddings
>
>
>
> [2] Observed Versus Latent Features for Knowledge Base and Text Inference
>
> ## Modality Data and Encoder Details
> For the modality data in each dataset, we follow the setting of their original paper and use their released multi-modal features. We present their dimensions in Table 1 and further list the pre-trained backbone used for each dataset in the following table:
>
> | Dataset | Image Backbone | Text Backbone |
> | :---: | :---: | :---: |
> | MKG-W | BEiT | Sentence-BERT |
> | MKG-Y | BEiT | Sentence-BERT |
> | DB15K | VGG | BERT |
> | KVC16K | ViT | BERT |
>
>
> This information could be also found in the original papers. Note that in the paper, we just mention BERT and VGG as an **examples** of the more diverse backbones that will be used in the actual dataset, and to ensure a fair comparison of the experimental results, we use the same modality features across baselines. When handling multiple images, the entity visual embedding would be an average of multiple image features.
>
> ## Notation Standardization
> We will continue to optimize the formula notations in our paper. In formula (1), the notation $ \mathcal{E}/\{h\} $represents the entity set without h. It is not $ \mathcal{E}_{h} $ . Perhaps there is some **misunderstanding**. We have changed this symbol to a "minus sign (-)" for easier understanding.
>
> ## About the temperature
> The relation-wise temperature is a learnable scalar parameter for each relation. We first initialize them randomly and they can be optimized during training. When calculating the weights in Equation (2), each relation-wise temperature is used with a sigmoid function to limit its range to (0, 1). By doing this, we can adjust the weights of each expert with guidance from relations.
>
>
>
> ## Score Function Comparision
>
>
> Different MMKGC baseline methods have their specific designs, and we have selected as many recent SOTA methods as possible for comparison in Table 2 of the main experiments, and also provided the performance of the original TuckER score, and from the comparisons of these results, we can find that we are in the Tucker's performance has made a significant improvement. This suggests that the improvement we achieved is based on better utilization of modal information. This is also evidenced in the ablation study.
>
>
>
> ## More analysis and result about the ablation studies (weakness 7 & 8)
> We found this comment so valuable that we did not put this part of the results in the initial manuscript because of typographical issues. Now we put the experimental results of MKG-Y and KVC16K in the list below and do a more in-depth analysis of the whole ablation study.
>
> | **Setting** | **MKG-Y** | | | |
> | :---: | :---: | --- | --- | --- |
> | | **MRR** | **Hit1** | **Hit3** | **Hit10** |
> | **Full Model** | **37.91** | **35.09** | **39.20** | **43.20** |
> | **(1.1) Structural Modality** | 35.66 | 32.14 | 37.61 | 41.51 |
> | **(1.2) Image Modality** | 35.99 | 32.84 | 37.98 | 41.23 |
> | **(1.3) Text Modality** | 35.73 | 32.41 | 37.76 | 41.18 |
> | **(1.4) Joint Modality** | 35.83 | 32.14 | 37.81 | 42.60 |
> | **(2.1) w/o relational temperature** | 37.17 | 33.93 | 38.66 | 42.85 |
> | **(2.2) noise** | 37.63 | 34.62 | 39.11 | 43.11 |
> | **(2.3) w/o adaptive fusion** | 36.54 | 33.05 | 38.17 | 42.28 |
> | **(2.4) w/o joint training** | 35.33 | 31.69 | 37.01 | 40.99 |
> | **(2.5) w/o ExID** | 36.37 | 32.56 | 38.64 | 42.70 |
>
>
>
>
> | **Setting** | **KVC16K** | | | |
> | :---: | :---: | --- | --- | --- |
> | | **MRR** | **Hit1** | **Hit3** | **Hit10** |
> | **Full Model** | **16.87** | **10.53** | **18.26** | **29.20** |
> | **(1.1) Structural Modality** | 15.66 | 9.52 | 16.95 | 27.54 |
> | **(1.2) Image Modality** | 15.50 | 9.56 | 16.59 | 27.11 |
> | **(1.3) Text Modality** | 15.56 | 9.61 | 16.68 | 27.16 |
> | **(1.4) Joint Modality** | 16.45 | 10.22 | 17.61 | 28.59 |
> | **(2.1) w/o relational temperature** | 16.74 | 10.31 | 18.16 | 29.01 |
> | **(2.2) noise** | 16.77 | 10.40 | 18.08 | 29.07 |
> | **(2.3) w/o adaptive fusion** | 16.55 | 10.28 | 17.69 | 28.70 |
> | **(2.4) w/o joint training** | 15.92 | 9.69 | 17.20 | 27.91 |
> | **(2.5) w/o ExID** | 16.05 | 9.87 | 17.24 | 28.05 |

---

> ### Author Response · Authors · 2024-11-18
> **Official Rebuttal from Authors (3)**
>
> These experimental results are consistent with those shown in the paper for MKG-W and DB15K. That is, each module contributes to the final performance. Besides, we can also do some analysis from the new experimental results on this issue of the small impact of the different modules that you mentioned. On the one hand, we conducted a fine-grained ablation study, with specialized experiments on several key designs. On the other hand, we can observe that there is also a big difference in the contribution of the same module in different datasets. This is related to the dataset itself, in addition to the fact that the foundational elements of our entire framework, such as the TuckER score itself, have better performance and subsequent designs have made significant enhancements based on this foundation.
>
>
>
> These are our responses to your comments and questions, and we hope that our responses will answer your concerns and improve your impression of our paper. We hope that you can improve the score of your decision, which will mean a lot to us.

---

### Author Response · Authors · 2024-11-18
**A Gentle Message for ICLR Rebuttal from Authors**

Dear PCs, SACs, ACs, and Reviewers:

After many days of hard work, we have finished **responding to each question in the review comments one by one**, and for some questions, we have added the corresponding experimental results. We hope you can see our Rebuttal responses and give us your feedback. We hope that our responses can solve your problems and improve your impression of our paper.

Sincerely,

Authors

---

> ### Author Response · Authors · 2024-11-18
> **Revised Version Uploaded**
>
> Dear PCs, SACs, ACs, and Reviewers:
>
> We have also uploaded the revised version of our paper after fixing the typos and adding more contents in it.
>
> Sincerely,
>
> Authors

---

### Author Response · Authors · 2024-11-21
**A Gentle Request for Rebuttal**

Dear Reviewers:

As the discussion deadline getting closer, we gently hope that you can participate in the rebuttal and give some possible feedback on our rebuttal responses.

Sincerely,

Authors

---

### Author Response · Authors · 2024-11-23
**A Gentle Request for Rebuttal**

Dear Reviewers:

As the rebuttal period would be end. We really hope that you can read our response and give us some further feedback.

Sincerely,

Authors

---

> ### Author Response · Authors · 2024-11-26
> **A Gentle Request for Rebuttal**
>
> Dear Reviewers:
>
> We have now uploaded a new revision of our paper. In this revision, we add more experimental results mentioned in the review comments. We hope you can find time in your busy schedule to take a look. Your feedback is very important to us.
>
>
> Sincerely,
>
> Authors

---

### Meta-Review · Area_Chair_5nKX · 2024-12-18

**Metareview:**

This paper proposes a multi-modal knowledge graph (MMKG) representation learning model, a Mixture of Modality Knowledge experts (MoMoK for short), to learn adaptive multi-modal entity representations for better MMKG completion. A key design idea is to learn representations through relation guidance and expert information in decoupling MMKGs. The method exploits a relation-guided modal knowledge expert network, adaptively aggregating multi-view embeddings and adopting a multimodal joint decision mechanism.

The initial reviews raised concerns about missing metrics/datasets/ablation studies, limited novelty, and unclear advantages or details. The authors have worked to improve the quality of the paper during the rebuttal and discussion period by adding more experiments (e.g., showing additional metrics and conducting additional ablation studies) and providing more details and analyses. Some reviewers acknowledged this improvement. These additional experimental results and explanations should be reflected in the final version of the paper.

**Additional Comments On Reviewer Discussion:**

While the initial reviews raised some critical concerns, such as missing experiments and limited novelty, Reviewer Diph, Reviewer rUdt, and
Reviewer mwpc have responded to the authors' rebuttal and gave positive feedback. The additional experimental results and explanations the authors have provided seem to make the work more convincing. More details are provided in the metareview.

---

### Decision · Program_Chairs · 2025-01-22

Accept (Poster)